# One LR Doesn't Fit All: Heavy-Tail Guided Layerwise Learning Rates for LLMs

**Di He** [1 2 3] **Songjun Tu** [2 3] **Keyu Wang** [4] **Lu Yin** [5 †] **Shiwei Liu** [6 7 8 †]

## Abstract

Learning rate configuration is a fundamental aspect of modern deep learning. The prevailing practice of applying a uniform learning rate across all layers overlooks the structural heterogeneity of Transformers, potentially limiting their effectiveness as the backbone of Large Language Models (LLMs). In this paper, we introduce **Layerwise Learning Rate (`LLR`)**, an adaptive scheme that assigns distinct learning rates to individual Transformer layers. Our method is grounded in Heavy-Tailed Self-Regularization (HT-SR) theory, which characterizes the empirical spectral density (ESD) of weight correlation matrices to quantify heavy-tailedness. Layers with weaker heavy-tailedness are assigned larger learning rates to accelerate their training, while layers with stronger heavy-tailedness receive smaller learning rates. By tailoring learning rates in this manner, `LLR` promotes balanced training across layers, leading to faster convergence and improved generalization. Extensive experiments across architectures (from LLaMa to GPT-nano), optimizers (AdamW and Muon), and parameter scales (60M–3B, up to 100B tokens) demonstrate that `LLR` achieves up to 1.5× training speedup and outperforms baselines, notably raising average zero-shot accuracy from 47.09% to 49.02% for 1B models and from 48.58% to 50.61% for 3B models. A key advantage of `LLR` is its low tuning overhead: it transfers nearly optimal LR settings directly from the uniform baseline. Code is available at https://github.com/hed-ucas/Layer-wise-Learning-Rate.

[1]Shenzhen Institutes of Advanced Technology, Chinese Academy of Sciences [2]Peng Cheng Laboratory [3]University of Chinese Academy of Sciences [4]University of Tübingen [5]University of Surrey [6]Max Planck Institute for Intelligent Systems [7]ELLIS Institute Tübingen [8]Tübingen AI Center. Correspondence to: Lu Yin <l.yin@surrey.ac.uk>, Shiwei Liu <sliu@tue.ellis.eu>.

*Proceedings of the 43$^{rd}$ International Conference on Machine Learning*, Seoul, South Korea. PMLR 306, 2026. Copyright 2026 by the author(s).

## 1. Introduction

Learning rate (LR) configuration is a cornerstone in modern deep learning, shaping both the convergence dynamics of training and the generalization ability of the resulting models (LeCun et al., 2015). Despite the rapid evolution of the LLM era—including the rise of Transformers (Vaswani et al., 2017), self-supervised learning (Radford et al., 2018), chain-of-thought reasoning (Wei et al., 2022), and RLHF (Ouyang et al., 2022)—the predominant LR strategy has remained essentially unchanged: applying a single LR value across all layers, which we refer to as the *Uniform LR*.

This paradigm, however, was originally developed for architectures with relatively homogeneous designs, such as multi-layer perceptrons (MLPs) and convolutional neural networks (CNNs). We contend that it does not adequately capture the structural heterogeneity of Transformer-based LLMs, thereby constraining their convergence and potentially their final performance. Specifically, recent studies highlight the **architectural heterogeneity** of Transformers, where the Hessian spectra differ substantially across layer types (Zhang et al., 2024a; Wang et al., 2025; He et al., 2025). These findings suggest that applying a single uniform LR might be inherently suboptimal, motivating the need for layerwise learning-rate strategies that explicitly account for such architectural heterogeneity in Transformer-based LLMs.

Research endeavors have explored the layerwise LR. For instance, the ratio between the weight norm and gradient norm of each layer has been used to set layerwise LRs, with the goal of stabilizing large-batch training in CNNs (You et al., 2017) and BERT (You et al., 2019). More recently, Wang et al. (2025) identified sharpness disparities across Transformer modules and introduced a fixed layer-wise LR schedule determined via grid search. However, our preliminary evaluation in Figure 1 shows that these approaches are highly sensitive to LR tuning: their improvements emerge only when compared against the uniform baseline at its suboptimal LR, while they often underperform the uniform baseline when the latter is tuned to its optimal LR. This leaves an open research question: ***Does a principled layerwise LR scheme exist that can outperform the uniform LR even at its optimal LR?***

To this end, we propose **Layerwise Learning Rate (`LLR`)**

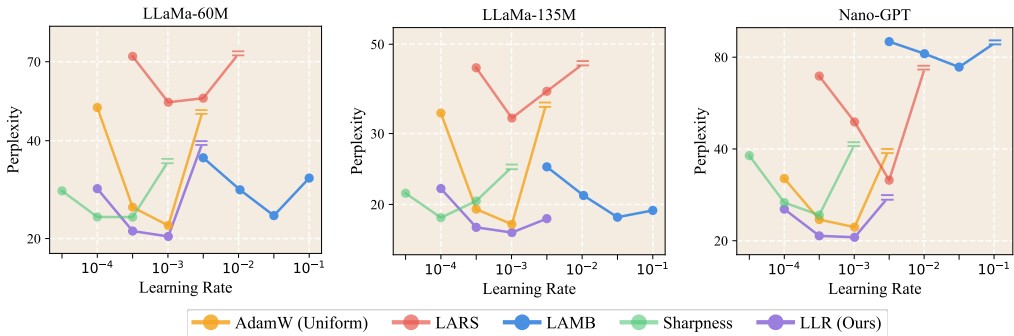

Figure 1. Learning rate sensitivity of different layerwise LR methods on LLaMa-60M, LLaMa-135M and Nano-GPT pre-training. Across all methods, data points failing to converge, resulting in excessively high perplexity values (exceeding the axis limits), are denoted by a **double dash ("=")**.

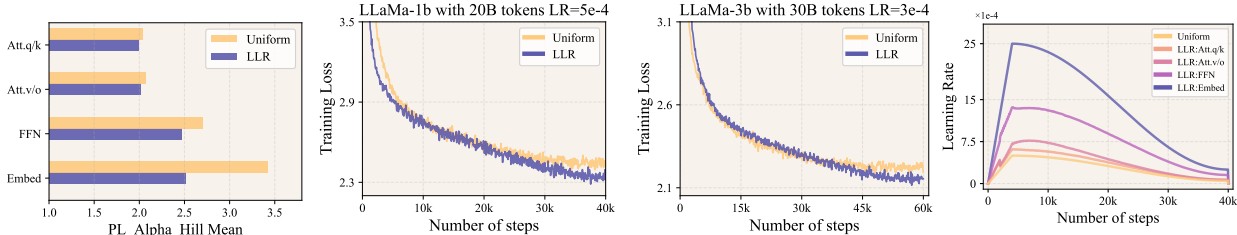

Figure 2. **Left:** Mean `PL_Alpha_Hill` value comparison (more balanced is preferred) between Uniform LR and `LLR`, with LLaMa-1B on FineWeb; **Middle:** Training loss curves of LLaMa-1B and LLaMa-3B under the AdamW optimizer; **Right:** Layerwise learning rate schedules when combining `LLR` with cosine decay scheduler. Performance comparison: `LLR` (Base LR=5e-4) obtains Perplexity (↓) =9.59 and Q-A Acc. (↑) =49.02%; `Uniform` (LR=5e-4) obtains Perplexity=9.77 and Acc.=47.09%, while `Uniform` (LR=2.5e-3) degrades to Perplexity=30.56.

for LLMs—an effective layerwise LR allocation strategy, grounded in Heavy-Tailed Self-Regularization (HT-SR) theory (Martin & Mahoney, 2019), that promotes balanced training across layers. `LLR` periodically measures the degree of heavy-tailedness in each layer's weight spectrum and assigns larger LRs to layers that are less heavy-tailed, while allocating smaller LRs to those exhibiting stronger heavy-tailedness. The underlying principle is drawn from HT-SR theory, which posits that layers with stronger heavy-tailedness are already better trained than those with weaker heavy-tailedness. Consequently, assigning larger LRs to the latter accelerates their training, thereby promoting more balanced progress across layers.

More concretely, `LLR` periodically computes the Empirical Spectral Density (ESD) across all layers, performs Power-Law (PL) fitting, and updates the corresponding `PL_Alpha_Hill` metrics. Layers with higher `PL_Alpha_Hill` values—typically embedding and FFN components—are assigned proportionally larger LRs, whereas layers with lower `PL_Alpha_Hill` values—such as attention modules—receive smaller LRs. This metric-to-map allocation scheme enables adaptive layerwise LR adjustment directly driven by spectral statistics, ensuring robust performance gains without extra hyperparameter-tuning burden. Our main contributions are as follows:

❶ We propose Layerwise Learning Rate (`LLR`), grounded by HT-SR theory, which enables dynamic assignment of LRs across layers within LLMs during training, promoting balanced training across layers.

❷ Extensive experiments on various architectures, including LLaMa to GPT-nano, optimizers with AdamW and Muon, from 60M to 3B, show that `LLR` yields up to a 1.5× training speedup (Figure 8). Under the same training token budget, `LLR` further surpasses existing layerwise approaches by a clear margin (Table 2 to Table 4).

❸ A key advantage of `LLR` is its low tuning overhead: it inherits nearly optimal LR settings directly from the standard uniform LR (Figure 1), thereby lowering the practical barrier to adoption.

## 2. Related Work

**Layerwise Learning Rate.** Although a uniform LR is the standard in deep network training, recent work has investigated layerwise LR allocation to improve optimization efficiency (Pan et al., 2024; Zhang et al., 2024b). LARS (You et al., 2017) and LAMB (You et al., 2019) scale LRs by the gradient-to-weight norm ratio, but as they were not designed for LLMs, they provide only marginal gains and demand additional tuning. Wang et al. (2025) introduced a blockwise LR allocation method that leverages sharpness

disparities to accelerate pretraining, though it relies on exhaustive grid search. Complementarily, Hayou & Liu (2025) observed a dependence of optimal embedding-layer LRs on vocabulary size, suggesting distinct scaling rules across components. The closest related work is TempBalance (Zhou et al., 2023), which adapts HT-SR to adjust learning rates at the layer level for improved optimization. However, its applicability is restricted to CNNs and fine-tuning scenarios with limited data volume.

**HT-SR theory for LLM**. HT-SR theory describes a statistical phenomenon in which the weights of well-trained neural networks exhibit strong correlations, giving rise to heavy-tailed patterns in the ESD of Layerwise weight matrices (Mahoney & Martin, 2019; Martin et al., 2021). Empirical evidence shows that, across various stages of training—whether at the early, intermediate, or final phase—different components of LLMs (Embed, Attention, FFN) display distinct heavy-tailed structures in their ESDs (Couillet & Liao, 2022; Kothapalli et al., 2025; Ba et al., 2022). Building on this observation, numerous studies have sought to balance these heavy-tailed characteristics through applications of HT-SR, including model selection (Mahoney & Martin, 2019; Martin et al., 2021; Yang et al., 2023), module-wise adaptive training (Zhou et al., 2023), LLM pruning (Lu et al., 2024), and module-wise weight-decay allocation (He et al., 2025), achieving notable improvements in large-scale model training. However, it remains unclear if HT-SR can be utilized for layerwise LR designing for LLM pre-training.

## 3. Methodology

In this section, we revisit the HT-SR theory and outline the key metrics that underpin our analytical framework. Subsequently, we investigate the spectral characteristics of weights in LLM pretraining tasks, and, informed by these findings, we introduce the LLR algorithm, which utilizes the HT-SR theory to deliver substantial improvements in pretraining performance.

### 3.1. HT-SR Theory

The HT-SR framework offers a systematic approach to characterizing the ESD of weight matrices in neural networks. Observations from prior work indicate that well-trained models tend to display ESDs with pronounced heavy-tails which is closely linked to higher training quality. Several studies have further revealed that, in LLMs, parameters that are well-optimized and those insufficiently trained can coexist throughout the entire training process, and such heterogeneity can significantly affect overall model performance (Zhou et al., 2023; He et al., 2025; Lu et al., 2024). Building upon this theoretical basis, we employ the HT-SR metric to measure the degree of spectral tail heaviness, applying lower LRs to layers with pronounced heavy-tail

characteristics (e.g., Att.q, Att.k) and higher LRs to those with weaker heavy-tails (e.g., FFN components, Embedding), thereby promoting balanced optimization across layers to enhance generalization and overall effectiveness (see Figure 2). The extent of heavy-tailedness is determined quantitatively by fitting a power law (PL) to the ESD and using the resulting PL exponent ($\alpha$) as the measurement criterion.

We consider a network comprising $N$ layers, each associated with a weight matrix $\{W_l\}_{l=1}^{L}$ of shape $n \times m$ ($n \leq m$). For each layer, the ESD is computed from the eigenvalues of the correlation matrix $X_l = W_l^\top W_l$. Formally, the ESD is defined as $\text{ESD}(x) = \frac{1}{N} \sum_{i=1}^{N} \delta(x - \sigma_i)$, where $\delta(\cdot)$ denotes the Dirac delta function and $\sigma_i$ are the singular values. We model the ESD by PL distribution of the form:

$$p(\lambda) \propto \lambda^{-\alpha}, \lambda_{\min} < \lambda < \lambda_{\max} \tag{1}$$

where $p(\lambda)$ represents the eigenvalue density within the specified range, and $\alpha$ serves as a quantitative measure of the heavy-tailedness of the spectrum. We visualize the ESD distributions and the corresponding PL fits in Figure 13 of Appendix C. The PL exponent $\alpha$ is estimated using the Hill estimator (Zhou et al., 2023; Liu et al., 2024). Let $\{\lambda_i\}_{i=1}^{n}$ denote the eigenvalues sorted in ascending order. The Hill estimator is given by:

$$\texttt{PL\_Alpha\_Hill} = 1 + \frac{k}{\sum_{i=1}^{k} ln \frac{\lambda_{n-i+1}}{\lambda_{n-k}}} \tag{2}$$

where the parameter $k$ controls the lower cutoff in the fitting process. In all experiments, we set $k = \frac{n}{2}$, thereby estimating the slope using the largest half of the eigenvalues (Detailed study of PL fitting method refer to Appendix B, Figure 10). In short, in the context of LLMs, a smaller $\alpha$ indicates higher trainability of the corresponding weight matrix.

### 3.2. Layerwise Learning Rate for LLMs (`LLR`)

Based on the `PL_Alpha_Hill` characteristics in Section 3.1, we propose a Layerwise LR (LLR) allocation method. LLR calculates the `PL_Alpha_Hill` values of all layers, assigning larger LRs to those with higher values and smaller LRs to those with lower values (see Figure 3). The mapping from `PL_Alpha_Hill` values to per-layer LRs follows the bounded scaling function:

$$f_T(i) = \eta \cdot \left( \frac{\alpha_T^i - \alpha_T^{\min}}{\alpha_T^{\max} - \alpha_T^{\min}} (s - 1) + 1 \right) \tag{3}$$

where $\eta$ is the global LR, and $(1, s)$ define the range of scaling ratios applied to $\eta$. $\alpha_T^i$ is the `PL_Alpha_Hill` value of layer $i$ at step $T \in \{0, \tilde{t}, 2\tilde{t}, ..., t_{max}\}$, with $\tilde{t}$ denotes the interval for calculating `PL_Alpha_Hill` and

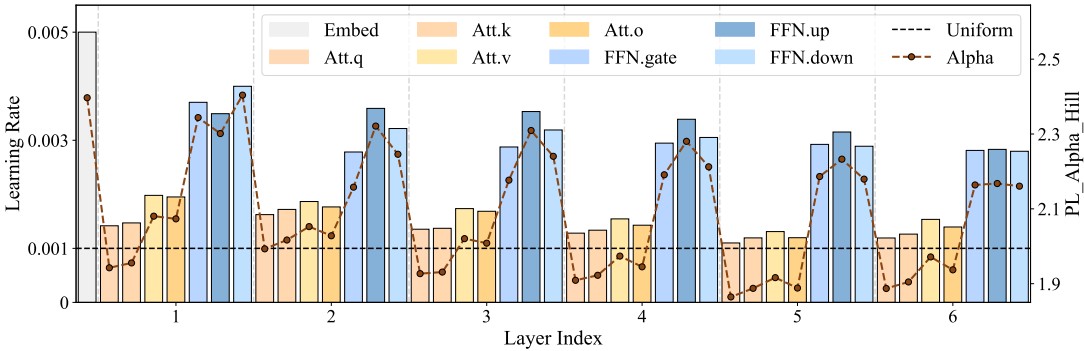

*Figure 3.* The bars denote the Learning Rate, and the line indicates the `PL_Alpha_Hill`. Given the imbalanced layerwise `PL_Alpha_Hill` of LLaMa-60M, `LLR` assigns lower LR to layers with lower `PL_Alpha_Hill`.

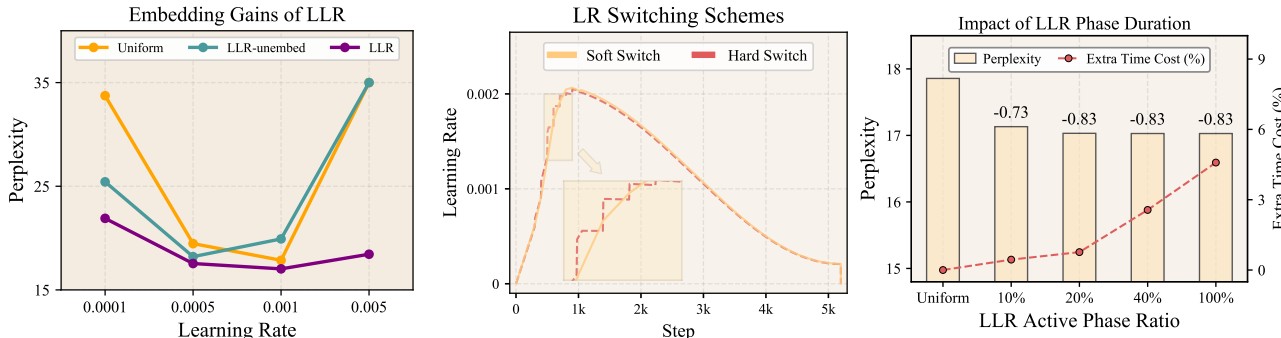

*Figure 4.* **Left:** Embedding gains of `LLR` compared to baselines across different LRs with LLaMa-135M. "LLR-unembed" denotes the ablation variant without the tailored spectral-based adjustment for Embedding layer; **Middle:** Illustration of hard (perplexity = 17.18) versus soft (perplexity = 17.03) LR switching schemes with LLaMa-135M; **Right:** Impact of `LLR` phase duration on perplexity and extra time cost with LLaMa-135M. The computation time for `Uniform` is 4.51 H100 hours.

$t_{max}$ the total training steps. While $\alpha_T^{min}$ and $\alpha_T^{max}$ are the minimum and maximum `PL_Alpha_Hill` values among all layers at step $T$. Formula (3) guarantees that the adjusted global LR, $f_T(i)$, remains within $[\eta, s\eta]$ as a scaled variant of $\eta$. Let $\eta^t$ and $f_T^t(i)$ denote the LRs at step $t \in \{0, 1, 2, ..., t_{max}\}$ under a cosine decay scheduler, corresponding to the global learning rate $\eta$ and the layer-wise value $f_T(i)$, respectively.

To effectively adapt the HT-SR theory to LLMs, we introduce three tailored designs targeting the unique characteristics of Transformer architectures:

**Tailored Embedding Treatment.** Considering the `Embedding` (and `Output_Head`) layer's persistently high `PL_Alpha_Hill` values across all training stages (refer to Figure 2 **(Left)** and Figure 14 of Appendix C), we fix its LR at the scaling function's upper bound $s\eta$, following Wang et al. (2025). This design is crucial to prevent the under-adaptation of the scaling function (Formula 3) that would otherwise occur. Figure 4 **(Left)** validates this approach: our tailored strategy yields significant performance gains compared to the **"LLR-unembed"** variant, ensuring that these critical layers receive sufficient updates.

**Soft Layer-wise LR Switch.** Traditional methods typically employ a **"Hard Switch"** that instantaneously shifts the LR $\eta^t$ to a target value $f_T^t(i)$, often causing detrimental **"LR spikes"** (as visualized in the red dashed line in Figure 4 **(Middle)**). To mitigate this, we propose a Soft Switch mechanism. Instead of an abrupt change, we linearly increase the current LR $\eta^t$ over a window of $t_{switch} \le \tilde{t}$ steps towards the target LR $f_T^{t+t_{switch}}(i)$ calculated at step $t + t_{switch}$. This "look-ahead" smoothing strategy eliminates spikes, thereby enhancing training stability. In our implementation, we set $t_{switch} = 0.5\tilde{t} = 50$ (approximately 2.6M tokens) by default (Detailed study refer to Appendix B, Figure 11).

**Efficient Active Phase.** We observe that the layer-wise `PL_Alpha_Hill` statistics tend to stabilize after processing approximately the first 20% of training tokens (as shown in Figure 5). Based on this finding, we restrict the computationally expensive layer-wise LR updates to this initial 20% phase. Figure 4 **(Right)** demonstrates that this strategy maintains comparable performance to full-time updates while drastically reducing computational overhead, making `LLR` highly efficient for large-scale pre-training.

We provide the details of `LLR` in Algorithm 1. By appropriately constraining dominant components, `LLR` establishes an adaptive and robust framework for layerwise LR allocation in LLM training.

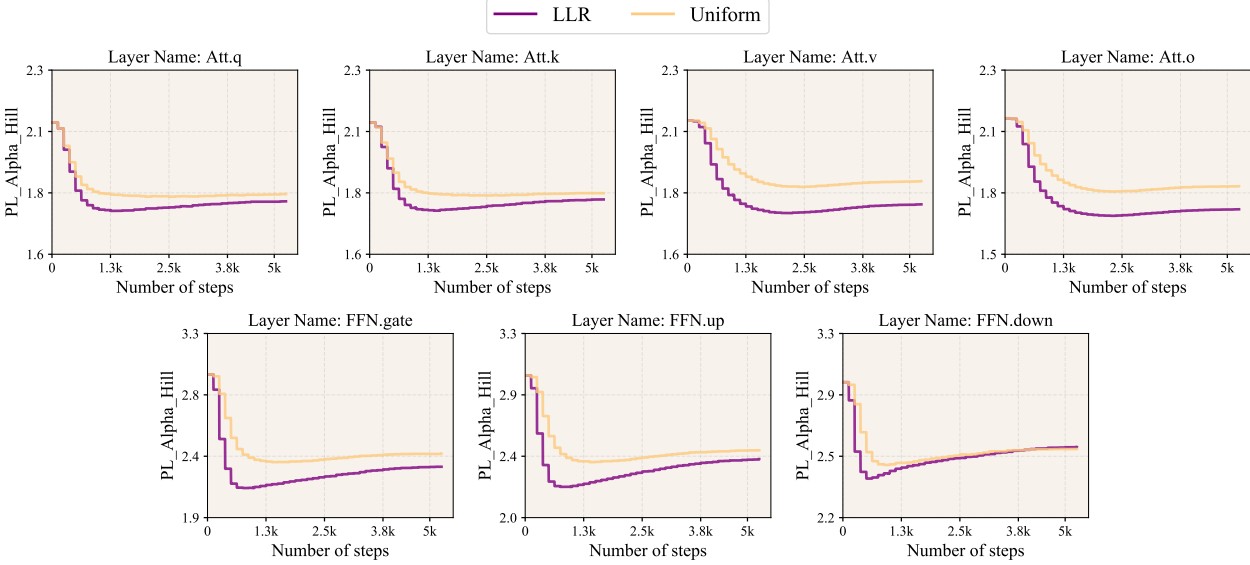

*Figure 5.* Evolution of `PL_Alpha_Hill` for different parameter groups when training LLaMa-135M with `LLR` (perplexity = 17.03) and `Uniform` (perplexity = 17.86). The curves are computed every 100 training steps.

---

**Algorithm 1** `LLR`

---

**Input:** Global LR $\eta$, number of training steps $t_{max}$, interval $\tilde{t}$ of using `LLR`, maximum scaling ratio $s$, switching steps $t_{switch}$, and $\alpha_T^i$ refers to $i_{th}$ layer's `PL_Alpha_Hill` at update step $T \in \{0, \tilde{t}, 2\tilde{t}, ..., t_{max}\}$

**for** $t \leftarrow 0$ **to** $t_{max}$ **do**
    **if** $mod(t, \tilde{t}) = 0$ **then**
        **Step 1:** Compute $\alpha_T^i$ for all layers using fomula (2);
        **Step 2:** Calculate the target global LR $f_T(i)$ using $\alpha_T^i$ and scaling function (3) bounded by $\eta$ and $s\eta$;
        **Step 3 (Soft Switch):** Linearly transition layer $i$'s LR from its current value to $f_T^{t+t_{switch}}(i)$ over the next $t_{switch}$ steps to avoid spikes;
        **Step 4:** Follow the cosine decay scheduler with global LR $f_T(i)$ starting from $f_T^{t+t_{switch}}(i)$ until $t + \tilde{t}$;
    **end if**
**end for**

---

### 3.3. Layer-wise `PL_Alpha_Hill` Dynamics

In this section, we investigate how `LLR` influences the optimization dynamics of different parameter groups, we analyze the evolution of the `PL_Alpha_Hill` in LLaMa-135M. Figure 5 shows the `PL_Alpha_Hill` for each major parameter group when training with `LLR` (perplexity = 17.03) and `Uniform` (perplexity = 17.86). We compute `PL_Alpha_Hill` every 100 update steps. Several trends are evident from the figure.❶ The `Attention` parameters (`Att.q`, `Att.k`, `Att.v`, `Att.o`) consistently exhibit smaller `PL_Alpha_Hill`, whereas the `FFN` parameters (`FFN.gate`, `FFN.up`, `FFN.down`) maintain larger `PL_Alpha_Hill` throughout training. ❷ The

`PL_Alpha_Hill` of all parameter groups change substantially during the early phase of training (approximately the first 20% of update steps). This pronounced variation highlights the importance of periodically updating the layer-wise LRs, rather than keeping them fixed over time. ❸ Compared with `Uniform`, `LLR` markedly reduces the `PL_Alpha_Hill` across all parameter groups. This reduction correlates with the improved perplexity achieved by `LLR`, suggesting that better control of `PL_Alpha_Hill` contributes directly to the enhanced training performance. For more detailed analysis of `PL_Alpha_Hill` about `LLR` and `Uniform`, please refer to Appendix C.

## 4. Empirical results

In this section, we begin by presenting the complete experimental setup (Section 4.1), followed by a comparison between `LLR` and several baselines (Section 4.2). Finally, we demonstrate the effectiveness of `LLR` across fine-tuning tasks, diverse model architectures, and various optimizers.

### 4.1. Experimental setup

**Models and Datasets**. We perform a comprehensive experimental study encompassing pre-training, zero-shot evaluation, and fine-tuning over a diverse range of model architectures and parameter scales. **(i) Pre-training.** Models are pre-trained on the FineWeb dataset (Penedo et al., 2024) under Chinchilla scaling law (Hoffmann et al., 2022). We evaluate two categories of models: LLaMa-based architectures (60M, 135M, 350M, 1B and 3B) and the GPT-nano architecture (135M). This design supports systematic evaluation of scalability and architectural generalization. **(ii) Zero-shot Commonsense Reasoning Evaluation.** Following pre-training, we evaluate the emergent reasoning capa-

*Table 1.* Hyperparameters used in pre-training experiments. `LLR-s` denotes the $(1, s)$ parameter setting in `LLR`.

| Model Size | Tokens | `LARS`-LR/WD | `LAMB`-LR/WD | `Sharpness`-LR/WD | `AdamW/LLR`-LR/WD | `LLR-s` |
|---|---|---|---|---|---|---|
| 60M | 1.5B | 0.001/1e-6 | 0.05/0.1 | 0.0005/0.1 | 0.001/0.1 | 5 |
| 135M | 3B | 0.001/1e-6 | 0.05/0.1 | 0.0001/0.1 | 0.001/0.1 | 5 |
| 350M | 7B | 0.001/1e-6 | 0.01/0.1 | 0.0001/0.1 | 0.001/0.1 | 5 |
| 1B | 20B | 0.001/1e-6 | 0.01/0.1 | 0.0001/0.1 | 0.0005/0.1 | 5 |

*Table 2.* (**Main result**). Comparison with `LLR` and all baselines on pre-training various sizes of LLaMa models on FineWeb dataset. Validation perplexity ($\downarrow$) is reported. All baselines are carefully tuned.

| Model Size | `Uniform` | LARS
(You et al., 2017) | LAMB
(You et al., 2019) | Sharpness
(Wang et al., 2025) | `LLR` |
|---|---|---|---|---|---|
| 60M | 21.94 | 52.52 | 23.53 | 23.28 | **20.30** |
| 135M | 17.86 | 32.78 | 18.59 | 18.54 | **17.03** |
| 350M | 12.96 | 19.04 | 14.56 | 14.91 | **12.71** |
| 1B | 9.77 | 11.81 | 10.68 | 9.77 | **9.59** |

*Table 3.* (**Zero-shot results of commonsense-reasoning tasks in Table 2**). Zero-shot evaluation results ($\uparrow$) on seven commonsense reasoning benchmarks using the LLaMa-1B model pretrained with different methods.

| Method | OBQA | Winogrande | ARC-c | ARC-e | Hellaswag | SIQA | PIQA | Avg. |
|---|---|---|---|---|---|---|---|---|
| `Uniform` | 28.0 | 55.01 | 30.72 | 66.50 | 39.29 | 40.38 | 69.75 | 47.09 |
| `LARS` | 24.0 | 52.25 | 27.05 | 58.71 | 33.36 | 38.18 | 67.36 | 42.99 |
| `LAMB` | 24.6 | 50.59 | 29.27 | 62.37 | 36.45 | 39.30 | 68.01 | 44.37 |
| `Sharpness` | 26.0 | 53.59 | 30.80 | 67.72 | 39.36 | 40.89 | 70.40 | 46.97 |
| `LLR` | **29.6** | **56.67** | **34.64** | **70.46** | **40.53** | 40.79 | 70.46 | **49.02** |

bilities of the LLaMa-1B and LLaMa-3B checkpoints. This is conducted via zero-shot inference on a comprehensive benchmark suite of seven unseen commonsense reasoning tasks: PIQA (Bisk et al., 2020), SIQA (Sap et al., 2019), HellaSwag (Zellers et al., 2019), WinoGrande (Sakaguchi et al., 2021), ARC-c (Clark et al., 2018), ARC-e (Clark et al., 2018), and OBQA (Mihaylov et al., 2018). The evaluation utilizes the standard prompts provided by the lm-evaluation-harness to quantify out-of-data performance. **(iii) Fine-tuning.** We fine-tune Llama-3.2-1B on the Commonsense Reasoning benchmark (Hu et al., 2023), which comprises the seven downstream tasks described above, enabling a direct comparison under a parameter-efficient adaptation setting.

**Baselines.** We compare with four representative methods: (i) `Uniform`: Based on standard AdamW (Loshchilov & Hutter, 2017) or Muon (Liu et al., 2025) optimizers, applies the same learning rate to all layers without any layer-wise differentiation; (ii) `LARS` (You et al., 2017): A first-moment-based optimizer that assigns per-layer learning rates proportional to the ratio of weight to gradient norms; (iii) `LAMB` (You et al., 2019): A second-moment-based adaptive optimization algorithm that scales learning rates in a layer-wise manner according to weight norms to ensure training stability; (iv) `Sharpness` (Wang et al., 2025): Determines a fixed learning rate ratio across different layers via grid search based on sharpness information, without providing

an automatic adjustment mechanism.

**Hyperparameters.** The detailed hyperparameter settings for all model sizes are summarized in Table 1 and Table 11 in Section A. All models are trained with gradient clipping at 1.0 and a cosine learning rate schedule, with 10% of the training tokens used for learning rate warmup. We conduct grid search over learning rates {0.00001, 0.00005, 0.0001, 0.0005, 0.001, 0.005} as shown in Figure 1. The best configuration for each scale are reported in the Table the corresponding $(1, s)$ parameter settings are also detailed in the tables. `LLR` is performed every 5.2M tokens (100 update steps) throughout all experiments (Detailed study refer to Appendix B, Figure 11).

### 4.2. LLM Pre-training

In this section, we present the experimental results for `LLR` and all baselines across architectures (from LLaMa to GPT-nano), optimizers (AdamW and Muon), and parameter scales (60M–3B), followed by a comprehensive comparison between `Uniform` and `LLR`.

#### 4.2.1. RESULTS UNDER CHINCHILLA SCALING LAW

Table 2 presents the effectiveness of `LLR` and all baselines on the pre-training of LLaMa models with varying parameter scales (60M, 135M, 350M, and 1B) on the FineWeb dataset under chinchilla scaling law.

*Table 4.* (**Large-scale experiments**). Valid perplexity (↓) and zero-shot evaluation results (↑) using the LLaMa-1B/3B model pretrained with 100B/30B tokens.

| Task | Method | Valid-Perplexity | OBQA | Winogrande | ARC-C | ARC-E | Hellaswag | SIQA | PIQA | Avg. |
|---|---|---|---|---|---|---|---|---|---|---|
| LLaMa-3B | Uniform | 9.02 | 27.4 | 55.56 | 33.36 | 70.16 | 42.31 | 39.92 | 71.33 | 48.58 |
| at 30B tokens | LLR | **8.86** | **29.4** | **59.04** | **34.39** | **72.43** | **43.81** | **42.68** | **72.52** | **50.61** |
| LLaMa-1B | Uniform | 8.70 | 30.8 | 57.93 | 37.37 | **73.4** | 44.48 | **41.71** | 72.42 | 51.16 |
| at 100B tokens | LLR | **8.67** | **32.2** | **60.38** | **38.4** | 72.35 | **45.24** | 41.42 | **73.01** | **51.86** |

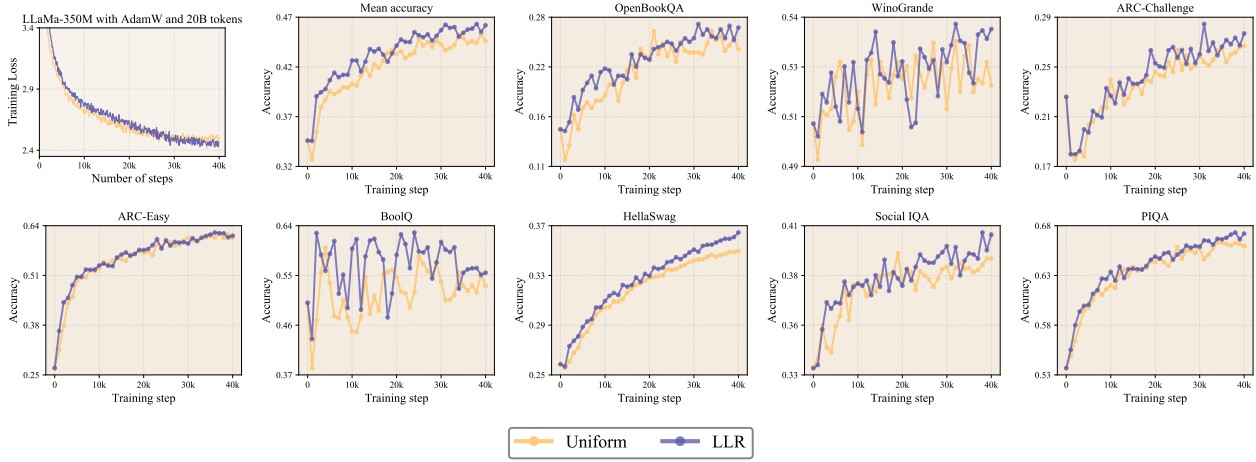

*Figure 6.* Training dynamics of LLaMa-350M with the AdamW optimizer over 20B tokens: training loss and zero-shot accuracy evaluated every 1,000 steps, including the average performance across eight downstream tasks. Figure 7 follows the same setting.

**Observations in Table 2.** ❶ **Superior and consistent gains across all baselines.** Across all evaluated model sizes, LLR surpasses both the Uniform baseline and the adaptive learning rate methods (LARS, LAMB, Sharpness). This consistent superiority over baselines demonstrates our approach's robustness for LLM pre-training. ❷ **Scalability to Larger Models.** The superiority of LLR is consistently observed from the smallest (60M) to the largest (1B) LLaMa models, achieving performance gains even at the billion-parameter scale.

Furthermore, our experiments reveal that existing methods, originally designed for architectures without Attention components, such as LARS and LAMB, do not yield optimal results for LLMs. This may be attributed to their lack of consideration for the distinct characteristics and optimization requirements of Attention and FFN layers within transformer architectures.

**Zero-shot Results in Table 3.** We evaluate the pretrained LLaMa-1B models, based on the checkpoints obtained from the pretraining experiments in Table 2, on 7 zero-shot commonsense reasoning tasks with lm-eval-harness under its default prompt configuration. As shown in Table 3, LLR achieves the best performance on all 7 benchmarks, boosting average accuracy from 47.09% to 49.02%. This confirms that pretraining gains effectively transfer to downstream reasoning tasks.

**LLaMa-3B with 30B tokens.** For the large-scale LLaMa-

3B experiment trained on 30B tokens, LLR consistently improves downstream zero-shot generalization. It outperforms the Uniform baseline on all 7 tasks and raises the average accuracy from 48.58% to 50.61%, demonstrating the effectiveness of layer-wise learning rate allocation at larger training scales.

### 4.2.2. RESULTS BEYOND CHINCHILLA SCALING LAW

**3x chinchilla: LLaMa-350M with 20B tokens.** Figure 6 and Figure 7 show training dynamics for LLaMa-350M under different optimizers over 20B tokens. LLR outperforms Uniform in terms of final average zero-shot accuracy: under AdamW, the accuracy improves from 44.50% with Uniform to 46.03% with LLR, and under Muon, it increases from 45.53% to 47.08%. Meanwhile, LLR also achieves slightly lower valid perplexity, reducing the loss from 11.29 to 11.05 with AdamW and from 11.28 to 11.08 with Muon. These results suggest that **LLR can substantially enhance the model's generalization capability on downstream zero-shot tasks** under both optimization settings.

**5x chinchilla: LLaMa-1B with 100B tokens.** To validate the effectiveness of LLR under extended training, we scaled the pre-training of LLaMa-1B from 20B to 100B tokens (Table 4). The results demonstrate that LLR maintains consistent performance gains even with a larger token budget. It outperforms the Uniform baseline in 5/7 tasks, improving the average zero-shot accuracy from 51.16% to 51.86%.

*Table 5.* Performance comparison with Layer-wise LR allocation methods. Training Performance of Different Optimization Methods across LLaMa Models of Varying Scales.

| Model Size | `AdamW` | `Adammini` (Zhang et al., 2024b) | `Mup-AdamW` (Yang & Hu, 2020) | `CompleteP-AdamW` (Dey et al., 2025) | `LLR` with `Mup-AdamW` | `LLR` |
|---|---|---|---|---|---|---|
| 60M | 21.94 | 21.85 | 24.52 | 24.26 | 22.54 | **20.30** |
| 135M | 17.86 | 17.72 | 18.46 | 18.15 | 17.54 | **17.03** |
| 350M | 12.96 | 13.04 | 13.56 | 13.42 | 13.28 | **12.71** |

### 4.2.3. MORE RELATED BASELINES

**`Adammini, mup and CompleteP`**. Table 5 shows `LLR` achieves the best results across all three LLaMa sizes, outperforming Adammini and other variants. This indicates that, compared with related layer-wise learning rate allocation strategies such as Adammini, Mup-AdamW, and CompleteP-AdamW, `LLR` achieves more consistent and scalable performance gains across different model sizes.

*Table 6.* Performance comparison with HT-SR based methods. We compare `LLR` against `Alphadecay` ((He et al., 2025)) and `Tempbalance` ((Zhou et al., 2023)) across 60M and 135M parameter scales.

| Model Siize | `Uniform` | `Alphadecay` | `Tempbalance` | `LLR` |
|---|---|---|---|---|
| 60M | 21.92 | 21.58 | 21.54 | **20.3** |
| 135M | 17.86 | 17.35 | 17.19 | **17.03** |

**`Alphadecay` and `Tempbalance`.** To isolate the contribution of our LLM-specific architectural designs, we benchmark `LLR` against two other approaches rooted in the same HT-SR theory: `Alphadecay` (He et al., 2025), which applies HT-SR to modulate weight decay, and `Tempbalance` (Zhou et al., 2023), which targets acceleration for simpler architectures like ResNet. As presented in Table 6, `LLR` achieves significantly lower perplexity compared to both baselines across different model scales (60M and 135M). This performance gap is instructive: while `Alphadecay` and `Tempbalance` share the theoretical underpinnings, they lack the tailored adaptations for the Transformer architecture. These results empirically validate our hypothesis that the specific designs introduced in Figure 4 **(e.g., [Tailored regularization for embedding layers, Soft layer-wise LR switch])** are essential for effectively translating the HT-SR theory into the LLM regime.

### 4.2.4. MORE BENEFITS FROM LLR

`LLR` demonstrates that **" one LR doesn't fit all "** and provides a substantial acceleration in model training.

**Performance of uniform LR at the scaling upper bound.** Table 7 presents the validation perplexity of different model sizes using AdamW optimizers with a uniform LR across all layers set to match the maximum Layerwise LR (upper bound) determined by `LLR`. Compared with the results in Table 2 and Table 7, `LLR` consistently outperforms both the upper bound and lower bound of uniform LR scaling

across different optimizers and model sizes. ***These findings confirm that a uniform LR configuration is inherently suboptimal***, whereas the proposed layer-wise LR strategy enables a more effective utilization of the capabilities of different optimizers.

*Table 7.* (**Up bound of LR scaling**). Validation perplexity (↓) of AdamW with a uniform LR equal to $s$ times that in `Uniform` of Table 2, matching the maximum LR from `LLR`'s scaling method.

| Model Size | 60M | 135M | 350M | 1B |
|---|---|---|---|---|
| LR (Upbound) | 0.005 | 0.005 | 0.005 | 0.0025 |
| `Uniform-Upbound` | 68.28 | 70.81 | 56.96 | 30.56 |
| `LLR` | **20.30** | **17.03** | **12.71** | **9.59** |

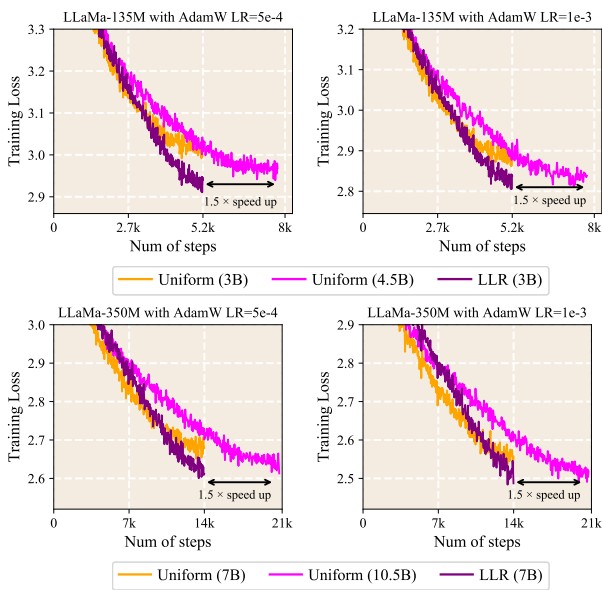

*Figure 8.* Training loss for the LLaMa-135M and LLaMa-350M model, comparing `Uniform` (3B/7B and 4.5B/10.5B tokens) against `LLR` (3B/7B tokens). The experiments employ AdamW under two different learning rates.

**Speedup.** Figure 8 presents the training loss curves for LLaMa-135M and LLaMa-350M model during pre-training, comparing `LLR` with `Uniform` across two learning rates. Across all configurations, `LLR` consistently achieves lower training loss with 3B/7B tokens compared to `Uniform` with the same number of training tokens, highlighting its effectiveness in accelerating convergence. Furthermore, `LLR` attains performance comparable to or better than that of `Uniform` at 4.5B/10.5B tokens, corresponding to an approximate 1.5× speedup.

*Table 8.* (**Finetuning tasks**). Finetuning results from the Commonsense Reasoning dataset using `Llama-3.2-1B`.

| Llama-3.2-1B | OBQA | Winogrande | ARC-c | ARC-e | Hellaswag | SIQA | PIQA | CSQA | Avg. |
|---|---|---|---|---|---|---|---|---|---|
| Un-tuned | 24.6 | 59.91 | 35.75 | 68.31 | 44.98 | 41.66 | 74.37 | 55.20 | 50.60 |
| Uniform | 25.0 | 64.80 | **38.82** | 68.14 | 45.29 | **47.54** | 72.25 | 61.67 | 52.94 |
| LLR | **25.4** | **65.02** | 38.64 | **68.64** | **45.49** | 46.54 | **76.25** | **62.67** | **53.58** |

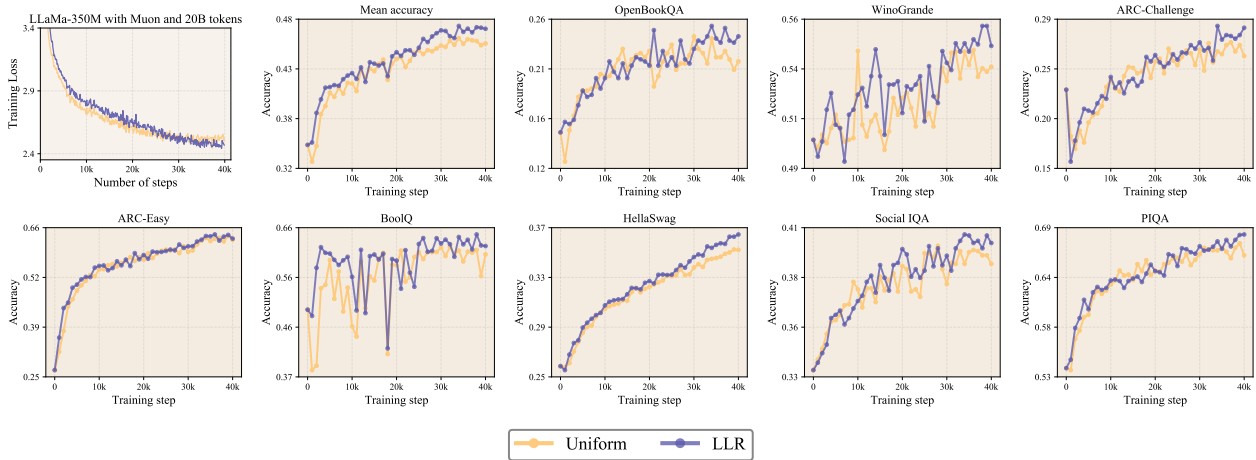

*Figure 7.* Training dynamics of LLaMa-350M with the Muon optimizer over 20B tokens.

### 4.3. More Analysis

In this section, we evaluate the downstream gains of all methods on finetuning tasks and the proposed `LLR` across more model architecture and optimizer to demonstrate its generality and robustness.

**Finetuning Results.** We evaluate all methods on finetuning tasks using `Llama-3.2-1B` from the Commonsense Reasoning dataset. As shown in Table 8, `LLR` achieves the best performance on 6/8 tasks. These results demonstrate that `LLR` is not only effective in the pretraining stage but also generalizes well to finetuning scenarios.

*Table 9.* (**GPT-nano Evaluation**). Performance comparison of `LLR` and all baselines on GPT-nano trained on the FineWeb dataset. Validation perplexity (↓) is reported.

| Method | Uniform | LARS | LAMB | Sharpness | LLR |
|---|---|---|---|---|---|
| Perplexity | 22.16 | 31.59 | 74.24 | 24.28 | **20.53** |

**Results of GPT-nano.** We evaluate the proposed `LLR` on GPT-nano, as shown in Table 9. The comparison includes `LARS`, `LAMB`, `Sharpness`, `Uniform` (AdamW), and `LLR`. The results demonstrate that `LLR` achieves the lowest validation perplexity, significantly outperforming all baseline methods. This further confirms the robustness and broad applicability of our approach across different model architectures.

**Results with Muon Optimizer.** We investigate whether the proposed `LLR` can consistently improve performance when paired with optimizers beyond AdamW. Specifically, Figure 7 and Table 10 presents results on various LLaMa model sizes trained with the Muon optimizer on the FineWeb dataset. Across all model scales, `LLR` delivers lower validation perplexity than `Uniform`, indicating that our method complements alternative optimizers like Muon, as its benefits stem from the spectral characteristics of the weight matrices rather than specific algorithmic mechanics.

*Table 10.* (**Pretraining with Muon**). Performance comparison on various sizes of LLaMa models trained with the Muon optimizer on the FineWeb dataset. Validation perplexity (↓) is reported. The reported LR is {LR for Muon part} and the {LR for AdamW part} is one-tenth of {LR for Muon part}.

| | LR | 0.05 | 0.01 | 0.005 | 0.001 |
|---|---|---|---|---|---|
| LLaMa-60M | Uniform | 20.02 | 19.49 | 21.07 | 32.32 |
| | LLR | **19.76** | **17.70** | **19.03** | **23.56** |
| | LR | 0.05 | 0.01 | 0.005 | 0.001 |
| LLaMa-135M | Uniform | 19.94 | 16.51 | 16.97 | 24.78 |
| | LLR | **19.24** | **16.29** | **16.00** | **18.24** |

## 5. Conclusion

We introduced `LLR`, a layer-wise learning rate adjustment strategy that leverages `PL_Alpha_Hill` during LLM training. Across diverse setups, including different model architectures, multiple optimizers, and varying parameter scales, `LLR` consistently delivers lower perplexity and better downstream performance than existing baselines, while maintaining robustness to variations in training configurations. Moreover, `LLR` accelerates convergence by up to 1.5× across multiple optimizers. More importantly, it inherits near-optimal learning rate settings from standard uniform LR, minimizing tuning overhead and enabling easy integration into existing pipelines.

## Acknowledgements

This work was supported by the Guangdong S&T Program (2024B0101010003).

## Impact Statement

This paper presents work whose goal is to advance the field of Machine Learning. There are many potential societal consequences of our work, none which we feel must be specifically highlighted here.

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

# Appendix

## A. Details of Experiments

This section provides detailed configurations for both pre-training and finetuning experiments. In Table 11 and Table 12, we present the architectural parameters of LLaMa models and the learning rate and weight decay settings for different methods of Nano-GPT.

*Table 11.* Hyperparameters of LLaMa models used in this paper.

| LLaMa-Size | 60M | 135M | 350M | 1B | 3B |
|---|---|---|---|---|---|
| Hidden | 768 | 768 | 1024 | 2048 | 2688 |
| Intermediate | 2048 | 2048 | 2736 | 5461 | 7680 |
| Heads | 6 | 12 | 16 | 32 | 32 |
| Layers | 6 | 12 | 24 | 24 | 28 |
| Batch Size | 512 | 512 | 512 | 512 | 512 |
| Sequence Length | 1024 | 1024 | 1024 | 1024 | 1024 |

*Table 12.* Learning Rate and Weight Decay used for different methods in pretraining GPT-nano on FineWeb dataset.

| Method | Uniform | LARS | LAMB | Sharpness | LLR |
|---|---|---|---|---|---|
| Learning Rate | 1e-3 | 5e-3 | 5e-2 | 1e-4 | 1e-3 |
| Weight Decay | 0.1 | 1e-6 | 0.1 | 0.1 | 0.1 |

## B. Ablation experiments

This section presents ablation experiments evaluating the effects of different LR scheduling strategies, metrics, and related factors.

**Repeat Experiments.** Table 13 provides a comparison of several LR scheduling strategies, evaluated through repeated experiments with different random seeds. The dependent t-test results further substantiate these findings, with statistically significant p-values supporting the superiority of LLR over all other optimizers.

*Table 13.* (**Dependent t-test results on FineWeb with LLaMa-135M using the AdamW optimizer**). Each method is evaluated over six repeated experiments with random seeds $\{5, 6, 7, 8, 9, 10\}$, and compared against LLR using a dependent t-test. Perplexity is reported as mean $\pm$ standard deviation. The resulting p-values correspond to comparisons with LLR.

| Method | Uniform | LARS | LAMB | Sharpness | LLR |
|---|---|---|---|---|---|
| Perplexity | 17.79 ($\pm$ 0.05) | 17.07 ($\pm$0.04) | 33.06 ($\pm$0.42) | 18.58 ($\pm$0.15) | **18.84** ($\pm$**0.25**) |
| P-value | 1.0e-10 | 1.9e-9 | 6.3e-7 | 9.3e-6 | |

*Table 14.* (**Inver projection of LLR**). Effect of Inverting the LLR Projection on Model Perplexity.

| Model | Uniform | Random | LLR-inv | LLR |
|---|---|---|---|---|
| 135M | 17.86 | 17.97 | 27.18 | **17.03** |
| 350M | 12.96 | 13.12 | 20.98 | **12.71** |
| 1B | 9.77 | 9.82 | 15.64 | **9.59** |

**Inver projection of LLR.** Table 14 evaluates the effect of reversing the projection used in LLR. Across different model scales, LLR achieves the lowest PPL, outperforming Uniform and substantially outperforming LLR-inv. In contrast, LLR-inv leads to clear performance degradation, indicating that the original LLR projection is properly aligned and critical for model performance.

*Table 15.* (**Attention-only Structure**). Validation Perplexity of the Attention-only 135M Model under Different LRs.

| Only-Att | 0.001 | 0.0008 | 0.0005 | 0.0003 |
|---|---|---|---|---|
| Uniform | 22.80 | 23.23 | 24.8 | 27.98 |
| LLR | **22.23** | **22.31** | **23.27** | **24.62** |

**Attention-only Structure.** Table 15 evaluates a modified 135M architecture from Table 11, where FFN modules are removed and only attention blocks are retained, to assess LLR on a more homogeneous structure. Across all learning rates, LLR consistently achieves lower PPL than Uniform, reducing PPL from 22.80 to 22.23, 23.23 to 22.31, 24.80 to 23.27, and 27.98 to 24.62. These results indicate that **LLR remains effective even in homogeneous attention-only architectures**.

**WSD scheduler.** Table 16 reports the validation PPL of LLaMa-60M, 135M, and 350M trained with the WSD scheduler. LLR consistently outperforms the Uniform baseline, reducing PPL from 21.73 to 21.01, 17.64 to 17.16, and 13.06 to 12.81, respectively. This indicates that LLR remains effective across different model scales under the WSD scheduler.

*Table 16.* (**WSD scheduler**). Validation Perplexity under the WSD Scheduler across Different Model Scales.

| WSD | 60M | 135M | 350M |
|---|---|---|---|
| LR | 0.001 | 0.001 | 0.0006 |
| Uniform | 21.73 | 17.64 | 13.06 |
| LLR | **21.01** | **17.16** | **12.81** |

*Table 17.* (**ViT-Tiny**). Top-1 Accuracy of ViT-Tiny under Different Training Methods.

| Backbone/Dataset | Metric | Uniform | LLR |
|---|---|---|---|
| ViT-Tiny/Imagenet-1K | Top-1↑ | 67.42 | 68.51 |

**ViT-Tiny.** Table 17 compares the Top-1 accuracy of ViT-Tiny trained with Uniform and LLR. LLR achieves a higher accuracy than Uniform, improving Top-1 accuracy from 67.42% to 68.51%, indicating that LLR provides better optimization performance for ViT-Tiny.

**Varying LR assignment functions.** We examine the performance of PL_Alpha_Hill with different LR assignment functions, which determine the allocation ratios of LR across different modules. Table 18 presents the results

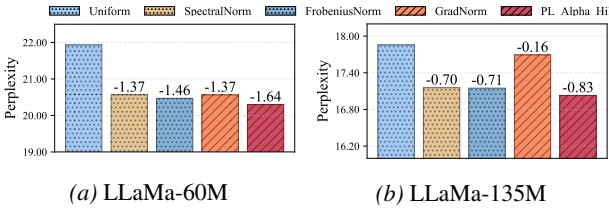

*(a)* LLaMa-60M      *(b)* LLaMa-135M

*Figure 9.* **(Varying HT-SR metrics).** Comparing `PL_Alpha_Hill` with multiple metrics under different learning rate settings. The value on the top of each bar indicates the difference from the leftmost bar in each plot and the same processing is applied in Figure 10, Figure 11 and Figure 12.

obtained by different assignment functions: Uniform, Linear, Sqrt and Log2. Among these, Linear achieves the best results across all LR settings, showing a notable advantage over other methods.

*Table 18.* **(Varying LR assignment functions).** Results of using different LR assignment functions under different LR settings. All experiments are conducted on LLaMa-60M.

| LR | Uniform | Linear (Ours) | Sqrt | Log2 |
|---|---|---|---|---|
| 0.001 | 17.86 | **17.02** | 17.16 | 17.14 |
| 0.0005 | 19.47 | **17.55** | 17.94 | 17.86 |

- **Sqrt:**

$$f_t(i) = \eta \frac{\sqrt{\alpha_t^i}}{\frac{1}{L}\sum_{j=1}^L \sqrt{\alpha_t^j}}$$

- **Log2:**

$$f_t(i) = \eta \frac{\log_2(\alpha_t^i)}{\frac{1}{L}\sum_{j=1}^L \log_2(\alpha_t^j)}$$

Here, $\eta$ denotes the initial LR, $\alpha_t^i$ is `PL_Alpha_Hill` of the module $i$ at step $t$, and $L$ is the total number of model layers.

**Varying HT-SR metrics.** To investigate the effect of different learning rate scheduling metrics on model performance, we conducted ablation studies comparing these methods with LLaMa-60M and LLaMa-135M. While prior work has primarily explored `Uniform` scheduling and `GradNorm` as representative metrics, our study additionally evaluates `PL_Alpha_Hill`, `FrobeniusNorm`, and `SpectralNorm` under identical training settings. Results in Figure 9 show that, `PL_Alpha_Hill` consistently achieves the lowest validation perplexity (lower is better), highlighting its effectiveness over other evaluated metrics.

**Varying PL fitting methods.** In our proposed framework, the HT-SR metric `PL_Alpha_Hill` is derived through PL fitting, and the choice of fitting method can affect final training effectiveness. To assess this impact, we evaluate `Goodness-of-fit` (Alstott et al., 2014; Martin et al., 2021; Clauset et al., 2009), `Fix-finger` (Yang et al.,

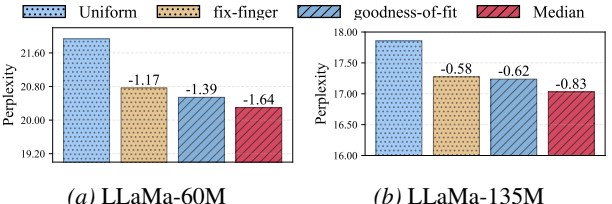

*(a)* LLaMa-60M      *(b)* LLaMa-135M

*Figure 10.* **(Varying PL fitting methods).** Analysis of three PL fitting methods—`Goodness-of-fit`, `Fix-finger`, and `Median`—across LLaMa-60M and LLaMa-135M.

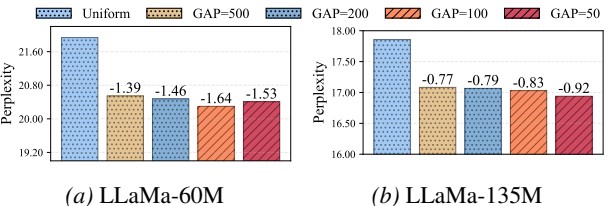

*(a)* LLaMa-60M      *(b)* LLaMa-135M

*Figure 11.* **(Varying PL fitting gaps).** PL fitting is conducted at varying gaps across training steps to evaluate the trade-off between performance. Bars represent validation perplexity (lower is better).

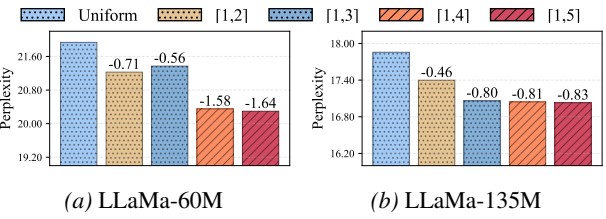

*(a)* LLaMa-60M      *(b)* LLaMa-135M

*Figure 12.* **(Hyperparameter study on** $(1, s)$**).** Results of a hyperparameter search for $(1, s)$ across LLaMa-60M and LLaMa-135M on the FineWeb dataset. The bar plots display validation perplexity (lower is better).

2023), and `Median` (Zhou et al., 2023) under identical experimental conditions in Figure 10. Across all tested experiments, `Median` maintains competitive or superior training performance compared to the other two approaches, making it the preferred choice for PL fitting within our method.

**Varying PL fitting gaps.** To evaluate the effect of PL fitting frequency on model performance, we compare different update gaps under identical experimental conditions in Figure 11. Across all tests, our method achieves stable performance across all gap settings, consistently outperforming the uniform baseline. Even when the fitting interval is as large as 500 training steps, the method maintains competitive perplexity, demonstrating robustness. These results justify using a larger fitting gap in practice to balance performance.

**Hyperparameter study on** $(1, s)$**.** Figure 12 demonstrates that `LLR`, with $(1, s)$ settings of $(1, 2)$, $(1, 3)$, $(1, 4)$ and $(1, 5)$, consistently surpasses the `Uniform` baseline across all experiments, with stable gains across schedules, highlighting its robustness and insensitivity to reasonable variations in $s$.

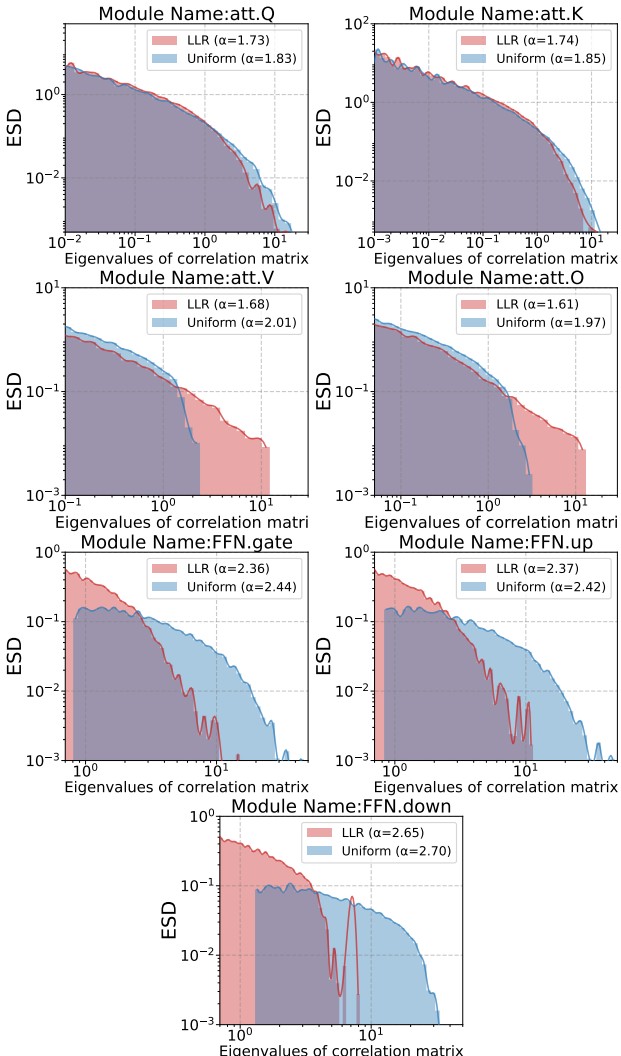

*Figure 13.* Comparison of ESD distributions across layers of LLaMa-135M under different training methods (`LLR`: Perplexity=17.03 vs. Uniform: Perplexity=17.86). Attention-related layers (e.g., att.q, att.k) exhibit notably heavier spectral tails in contrast to FFN-associated layers. Our method systematically balances the heavy-tailed properties across layers by appropriately configuring layer-wise LR, thereby enhancing overall model performance.

## C. Spectral Feature Analysis of `LLR`

To understand the structural impact of our proposed method on the model weights, we analyze the ESD plot of the layer-wise correlation matrices. Figure 13 presents a comparative visualization of the spectral distributions for the LLaMa-135M model trained with `LLR` versus the Uniform baseline.

The plots illustrate the distribution of eigenvalues on a log-log scale, where the slope of the tail relates to the PL exponent $\alpha$. A smaller $\alpha$ indicates a "heavier" tail, which is theoretically associated with better generalization capabilities in deep networks.

Comparing the two methods, we observe distinct spectral

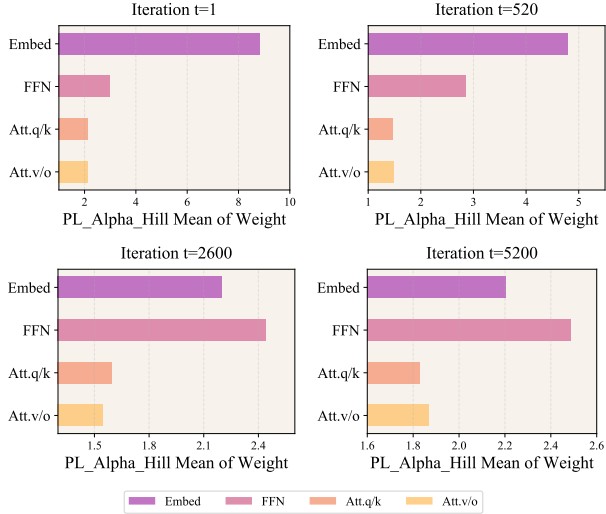

*Figure 14.* Evolution of `PL_Alpha_Hill` distributions from weights of LLaMa-135M components across iterations during pre-training on the Fineweb dataset, trained by AdamW with uniform LR.

behaviors:

- **Layer Heterogeneity:** Attention-related layers (e.g., `att.Q`, `att.K`) naturally exhibit heavier tails compared to FFN layers, suggesting different intrinsic regularization needs.

- **Effect of LLR:** The `LLR` method (shown in red) systematically shifts the spectral distributions compared to the Uniform baseline (shown in blue). For instance, in the `att.V` layer, `LLR` results in a heavier tail ($\alpha = 1.68$) compared to Uniform ($\alpha = 2.01$).

This modulation of spectral shapes demonstrates that `LLR` does not merely scale the weights but fundamentally alters the optimization trajectory, leading to a weight configuration that generalizes better (Perplexity: 17.03) than the Uniform approach (Perplexity: 17.86).

We further examine the evolution of spectral features across different components during the pre-training phase.

As illustrated in Figure 14, there is a significant heterogeneity in the `PL_Alpha_Hill` values among different layers. Specifically, attention-related components (e.g., `Att.q/k`, `Att.v/o`) consistently exhibit lower `PL_Alpha_Hill` values compared to FFN layers and Embeddings throughout the training iterations (from $t = 1$ to $t = 5200$). This persistent disparity indicates that different components have distinct capacities and learning dynamics, thereby justifying the necessity of the `LLR` strategy to assign layer-specific learning rates rather than a uniform global learning rate.

*Table 19.* Performance and Layer-wise Alpha Stability of `LLR` under Different LRs.

| Model | LR | 0.002 | 0.001 | 0.0007 | 0.0005 | 0.0003 |
|---|---|---|---|---|---|---|
| 135M | Uniform | 18.1(0.32) | 17.9(0.33) | 18.8(0.34) | 19.6(0.35) | 22.1(0.37) |
|  | LLR | 17.5(0.28) | 17.0(0.25) | 17.3(0.32) | 17.6(0.30) | 18.5(0.27) |
|  | **LR** | **0.001** | **0.0008** | **0.0006** | **0.0004** | **0.0002** |
| 350M | Uniform | 13.0(0.30) | 12.9(0.29) | 13.2(0.28) | 13.6(0.30) | 14.6(0.32) |
|  | LLR | 12.7(0.28) | 12.6(0.27) | 12.8(0.25) | 13.3(0.29) | 13.6(0.30) |

To analyze the effect of `LLR` on the inter-layer variability of alpha values, Table 19 report both PPL and the standard deviation of alpha across layers. Across all LR settings, `LLR` consistently achieves lower PPL and smaller layer-wise alpha standard deviations than Uniform, demonstrating improved performance, stability, and robustness. For larger-scale models, `LLR` also reduces PPL from 9.77 to 9.60 and alpha standard deviation from 0.28 to 0.25 on LLaMa-1B(refer to Table 2), and reduces PPL from 9.02 to 8.86 and alpha standard deviation from 0.29 to 0.26 on LLaMa-3B (refer to Table 4).

