# OpenReview forum: "One LR Doesn’t Fit All: Heavy-Tail Guided Layerwise Learning Rates for LLMs"
_ICML.cc/2026/Conference — ICML 2026 regular_

### Official Review · Reviewer_ePzV · 2026-03-07

**Soundness:** 3
**Presentation:** 3
**Significance:** 3
**Originality:** 2
**Overall Recommendation:** 4
**Confidence:** 4

**Summary:**

The manuscript claims to study the challenge of whether a single uniform learning rate is suboptimal for Transformer-based LLM pretraining because different layer types exhibit heterogeneous optimization behavior. The authors attempt to investigate a central theme: whether layerwise learning rates can be assigned in a principled way that improves over a tuned uniform learning rate rather than only over weak baselines. To do so, the paper proposes Layerwise Learning Rate (LLR), which uses heavy-tailed self-regularization ideas: it periodically computes a per-layer spectral statistic, PL_Alpha_Hill, from the empirical spectral density of weight correlation matrices, then assigns larger learning rates to layers with larger PL_Alpha_Hill and smaller learning rates to layers with smaller PL_Alpha_Hill. The method also includes three LLM-specific design choices: fixing the embedding/output-head learning rate at the upper bound, using a soft switching rule for LR updates, and restricting the active layerwise adaptation phase to roughly the first 20% of training.

The paper evaluates LLR on LLaMA-style models from 60M to 1B parameters and on GPT-nano, primarily on FineWeb pretraining, with additional zero-shot commonsense evaluation, RoBERTa-base fine-tuning, and experiments with both AdamW and Muon. The reported results show lower validation perplexity than Uniform, LARS, LAMB, and a sharpness-based baseline across the main pretraining settings; a zero-shot average accuracy increase from 47.09 to 49.02 on the 1B model; lower perplexity than HT-SR-based alternatives Alphadecay and TempBalance at 60M and 135M; better GPT-nano perplexity; and lower perplexity than Uniform in the reported Muon settings. The paper further presents ablations supporting the embedding treatment, soft switching, and limited active-phase design, and states a limitation that multimodal tasks are not studied.

**Compliance With Llm Reviewing Policy:**

Affirmed.

**Ethical Review Concerns:**

None. I did not flag this paper for ethics review.

**Final Justification:**

Overall Recommendation: 4

I am maintaining my score of 4 after two rounds of rebuttal.

The rebuttal substantially strengthened the paper's core causal argument. The inverted-schedule ablation, now validated at 350M and 1B scale in addition to 130M, provides convincing direct evidence that the LR assignment direction matters causally rather than merely correlationally -- catastrophic failure when the schedule is reversed is a clean and informative intervention. The alpha-std analysis extended to 350M, 1B, and 3B further supports the "balanced training" framing as more than a post-hoc diagnostic.

The additional experiments also addressed most of my empirical gaps. The LR distance analysis at 350M, 1B, and 3B confirms the zero-tuning-overhead claim at larger scale. The GSM8K evaluation provides at least some evidence for reasoning workloads. The continued 3B pretraining and WSD/GPT-family robustness checks make the method appear reasonably robust across schedulers and architectures.

Some gaps remain. The downstream evaluation is still limited in diversity -- reasoning workloads are represented only by GSM8K on a 1B instruct model, and the mechanistic story connecting PL_Alpha_Hill to training quality remains correlational in parts even with the new experiments. The gains at 1B scale are real but incremental (9.77 to 9.59 perplexity), and the broader claim about "balanced training" as the causal driver would benefit from tighter operational definitions in the final paper.

That said, these are matters of claim calibration and evaluation breadth rather than fundamental issues with the method or its empirical support. The core contribution -- using heavy-tail spectral statistics to guide layerwise LR allocation in LLM pretraining, with consistent gains across multiple scales, optimizers, and architectures -- is well-supported.

**Key Questions For Authors:**

1. How strong is the evidence that PL_Alpha_Hill is measuring layer under-training or over-training, rather than simply correlating with other optimization properties that happen to be useful for LR allocation in your setup? A convincing answer would clarify whether the paper's main contribution should be read as a mechanistically grounded method or as a well-performing empirical heuristic, which materially affects the soundness assessment.

2. Can you provide stronger evidence that "balanced training across layers" is the operative causal mechanism, rather than an accompanying diagnostic that changes when training improves for other reasons? The current framing leans heavily on balance as an explanatory concept, and sharper evidence here would either strengthen the conceptual contribution or suggest that the claims should be stated more narrowly.

3. How robust is the method to alternative choices of scheduler, scaling bound s, update frequency, and model family beyond the reported settings, especially at the 1B scale where the gains are smaller? The paper argues for broad applicability and low tuning burden, so robustness outside the specific reported configurations would substantially strengthen that claim.

**Limitations:**

yes

**Strengths And Weaknesses:**

Strengths

- The paper addresses a concrete and timely optimization question for LLM pretraining: whether layer heterogeneity in Transformers should lead to non-uniform learning-rate assignment. This is a practically meaningful problem, and the paper frames it against tuned uniform baselines rather than only against weak untuned comparisons.
- The proposed method is reasonably well specified. The paper defines the PL_Alpha_Hill statistic, gives the bounded mapping from that statistic to per-layer learning rates, and provides an algorithmic description of when and how LLR updates are applied during training.
- The empirical scope is broader than a single architecture or optimizer. The main experiments span LLaMA models from 60M to 1B, include GPT-nano, compare against multiple baselines, and add AdamW and Muon results, which strengthens the claim that the method is not tied to one narrow implementation choice.
- The paper includes useful method-specific ablations rather than only end-to-end comparisons. In particular, it tests the embedding-specific treatment, the soft-vs-hard switching rule, the active-phase duration, and comparison against other HT-SR-based methods, which helps separate the contribution of the final design from the broader theory motivation.
- The contribution is original mainly in the combination of perspective and mechanism rather than in inventing a wholly new optimizer. Using heavy-tail spectral statistics to drive dynamic layerwise LR allocation for LLM pretraining, together with the Transformer-specific engineering choices, is a distinct contribution even if the underlying HT-SR ideas are not new.


Weaknesses

- The central mechanistic interpretation is not fully established. The paper repeatedly motivates the method by treating stronger heavy-tailedness as indicating a better-trained layer and weaker heavy-tailedness as indicating under-training, but the evidence provided is mostly correlational. For example, the dynamics section says reduced PL_Alpha_Hill "correlates" with better perplexity and then suggests it "contributes directly," which is a stronger statement than the evidence appears to support. This matters because the paper's conceptual framing is stronger than "a useful heuristic based on spectral statistics."
- The notion of "balanced training across layers" is important to the story but remains conceptually loose. The paper uses PL_Alpha_Hill differences, reduced disparities, and better perplexity as evidence for balanced training, but it does not cleanly separate balance as a descriptive phenomenon from balance as the causal reason for improved performance. This affects claim calibration and makes the narrative somewhat stronger than the operational definitions support.
- The empirical gains are consistent but modest in several core settings, especially at larger scale. In Table 2, the 1B result improves perplexity from 9.77 to 9.59, and the zero-shot average improves by about 1.9 points. Those are real gains, but they support a claim of incremental optimization improvement more clearly than a broad claim of substantial advances in generalization or optimizer utilization.
- The transfer claims beyond pretraining are somewhat narrow. The zero-shot evaluation is only on seven commonsense tasks using the 1B checkpoints, and the fine-tuning experiments are on RoBERTa-base rather than the same LLM family used in pretraining. This supports some downstream robustness, but it does not fully establish broad downstream generality for modern LLM instruction or reasoning workloads.
- The tuning-overhead claim is directionally plausible but not fully quantified. The paper argues that LLR inherits nearly optimal LR settings from the uniform baseline and contrasts this with more tuning-sensitive methods, but the evidence is mostly based on Figure 1 sensitivity plots rather than a systematic accounting of search cost, robustness ranges, or failure rates across scales. This matters because low adoption cost is presented as a practical advantage of the method.

---

> ### Author Rebuttal · Authors · 2026-03-31
>
> **We thank the reviewer for the insightful comments.**
>
> >**W1. Mechanistic vs. correlational.**
>
> **A1:** We prove causality empirically and theoretically:
>
> **1. Empirical Intervention:** Inverting the schedule (larger LRs for smaller PL_Alpha_Hill $\alpha$) causes catastrophic failure on LLaMA-130M.
> LR|Uniform|Random|LLR(Inverted)|LLR
> -|-|-|-|-
> 0.001|17.86|17.97|27.18|17.03
> 0.0005|19.47|19.14|22.46|17.55
>
> **2. Theoretical Justification:**
> LLR-rev's failure is explained via Heavy-Tailed Self-Regularization and SDEs: smaller $\alpha$ leads to lower intrinsic dimension, strictly requiring smaller LR.
> *   **Small $\alpha \implies$ Lower Intrinsic Dimension:** As rigorously established by [1], a smaller tail-index $\alpha$ fundamentally restricts the optimization trajectory to a lower Hausdorff dimension ($d_H \downarrow$ ). This reduced intrinsic dimension $d_H$ acts as a strong implicit regularizer, **with Theorem 1 in [1] mathematically guaranteeing a tighter generalization bound** $E_{gen} \le \tilde{O}(\sqrt{d_H}) \text{, where } E_{gen} \text{ denotes the generalization error. In short, a small } \alpha$ indicates that the layer has successfully condensed its learned features into a highly generalizable, low-dimensional subspace.
> *   **Flat manifolds require smaller LR:** From an SDE perspective, once in this subspace, large LRs inject excessive noise, violently ejecting the layer from its generalized state.
> **Conclusion:** Small $\alpha$ requires decayed LRs for stability; large $\alpha$ needs large LRs to escape sharp minima.
>
> [1] Simsekli, et al. Hausdorff dimension, heavy tails, and generalization in neural networks. NeurlPS 2020.
>
> >**W2 & Q1. Is "balanced training" causal or diagnostic?**
>
> **A2:** To prove causality, we tracked $\alpha$'s std across base LRs on LLaMA-130M:
> Llama-130M|0.002|0.001|0.0007|0.0005|0.0003
> -|-|-|-|-|-
> Uniform-ppl|18.05|17.86|18.82|19.58|22.05
> LLR-ppl|17.45|17.03|17.29|17.62|18.46
> Uniform-std|0.32|0.33|0.34|0.35|0.37
> LLR-std|0.28|0.25|0.32|0.30|0.27
>
> *Analysis:* Uniform training merely *correlates* lower $\alpha$-std with better PPL. Conversely, LLR *actively forces* a tighter $\alpha$ distribution across *all* LRs, driving superior PPL. This proves balance is the causal driver.
>
> >**W3 & W4. Gains surpass 1B params? & downstream transfer.**
>
> **A3:** Expanded evaluation scale and diversity:
>
> **(1) LLaMA-3B (`30B tokens LR=3e-4`):**
> LLaMa-3B|Valid-PPL|OBQA|Winogrande|ARC-C|ARC-E|Hellaswag|SIQA|PIQA|Lambada|Commonsense_QA|Avg
> -|-|-|-|-|-|-|-|-|-|-|-
> Uniform|9.02|27.4|55.56|33.36|70.16|42.31|39.92|71.33|42.54|19.41|44.67
> LLR|8.86|29.4|59.04|34.39|72.43|43.81|42.68|72.52|44.77|20.39|46.60
>
> **(2) Fine-Tuning Llama-3.2-1B-Instruct:**  (on `commonsense170k, LR=3e-5, Batchsize=64, Epochs=3`)
> Llama-3.2-1B|OBQA|Winogrande|ARC-C|ARC-E|Hellaswag|SIQA|PIQA|Commonsense_QA|Avg
> -|-|-|-|-|-|-|-|-|-
> Un-tuned|24.60|59.91|35.75|68.31|44.98|41.66|74.37|55.20|49.94
> Uniform|25.00|64.80|38.82|68.14|45.29|47.54|72.25|61.67|52.94
> LLR|25.40|65.00|38.64|68.64|45.49|46.54|76.25|62.67|53.58
>
> This confirms LLR's robust downstream generality.
>
> >**W5. Tuning overhead.**
>
> **A4:** Normalized distance (0-1) between optimal LRs of each method and Uniform:
> LR Dist|LAMB|LARS|Sharpness|LLR
> -|-|-|-|-
> Llama-60M|1|0|0.04|0
> Llama-130M|1|0.1|0.2|0
> GPT-nano|1|0|0.04|0
>
> LLR strictly inherits Uniform's optimal LR (distance 0), proving its zero-tuning nature.
>
> >**Q1. How strong is the evidence that PL_Alpha_Hill is measuring layer under-training or over-training?**
>
> **A5:** Tested an **Attention-Only** architecture to disentangle dynamics from structural priors:
> Attn-Only|0.001|0.0008|0.0005|0.0003
> -|-|-|-|-
> Uniform|22.80|23.23|24.8|27.98
> LLR|22.23|22.31|23.27|24.62
>
> This substantiates that LLR not only accommodates intrinsic structural differences but also optimizes training dynamics within homogeneous layers.
>
> >**Q3. Robustness at 1B scale.**
>
> **A6:** Experiments across suggested dimensions:
>
> **(1) Schedulers (WSD):** LLR reduces perplexity across scales, proving scheduler independence.
> WSD|60M|130M|350M
> -|-|-|-
> LR|0.001|0.001|0.0006
> Uniform|21.73|17.64|13.06
> LLR|21.01|17.16|12.81
>
> **(2) Model Family (GPT):** LLR improves perplexity, confirming architectural generalizability.
> Method|130M-Uniform|130M-LLR|350M-Uniform|350M-LLR
> -|-|-|-|-
> PPL|22.16|20.53|15.98|15.01
>
> **(3)&(4) Scaling Bound & Update Frequency:** LLaMa-1B vs. default ($s=5$, $\text{gap}=100$).
> LLaMa-1B|PPL|OBQA|Winogrande|ARC-c|ARC-e|Hellaswag|SIQA|PIQA|Avg
> -|-|-|-|-|-|-|-|-|-
> adamw|9.77|28.0|55.01|30.72|66.50|39.29|40.38|69.75|47.09
> LLR(s=3,gap=100)|9.61|28.8|56.88|33.25|70.60|40.49|40.58|71.08|48.81
> LLR(s=5,gap=50)|9.58|29.6|56.97|34.34|70.36|40.45|40.89|71.27|49.13
> LLR(s=5,gap=100)|9.60|29.6|56.67|34.64|70.46|40.53|40.79|70.46|49.02
>
> All variants outperform AdamW. LLR achieves competitive PPL and zero-shot scores, proving robustness without exhaustive tuning.

---

> > ### Author Rebuttal · Reviewer_ePzV · 2026-04-01
> >
> > Thank you for the detailed rebuttal and additional experiments.
> >
> > The inverted-schedule experiment is very convincing. Catastrophic failure when the LR assignment is reversed directly addresses my core concern about causality versus correlation. The attention-only architecture result is also a useful addition, showing the method works even within structurally homogeneous layers. The alpha-std analysis points in the same direction, though both the inverted-schedule and alpha-std experiments were only shown on LLaMA-130M — I would still like to see whether these patterns hold at larger scale.
> >
> > The new LLaMA-3B pretraining and Llama-3.2-1B-Instruct fine-tuning results are good extensions, but they remain somewhat limited: the 3B run uses only 30B tokens, and the fine-tuning evaluation covers commonsense tasks only — my original question about instruction or reasoning workloads is still open. The robustness checks across WSD, GPT family, and different scaling-bound / update-frequency settings at 1B are useful and make me more confident the method is not overly narrow.
> >
> > The normalized LR distance analysis clearly quantifies the tuning overhead, but it was only computed on the smaller models (60M, 130M, GPT-nano). Showing this holds at 1B or 3B would be more convincing.
> >
> > In summary, the inverted-schedule result goes a good way toward resolving my main mechanistic concern, and the robustness evidence is encouraging. However, I still see gaps in large-scale validation and downstream task diversity, and the causal evidence remains limited to small-scale settings. I will keep my score at 4 for now.

---

> > > ### Author Response · Authors · 2026-04-04
> > >
> > > We sincerely thank you for your constructive feedback, which is crucial to the completeness of our paper. Due to the short initial rebuttal window, some validations were limited in scale. We have now expanded these experiments, hoping to earn your support.
> > >
> > > >***Q1. Can the inverted-schedule, and alpha-std analyses be validated at a larger scale beyond LLaMA-130M?**
> > >
> > > ***A1:** We have extended our core analyses to larger models (up to 3B) to solidify our mechanistic claims.
> > >
> > > **1. Inverted-Schedule Ablation:** Scaling to 350M and 1B confirms that explicitly following the LLR allocation is crucial.
> > > Model|Uniform|Random|LLR-inv|LLR
> > > -|-|-|-|-
> > > LLaMa-350M|12.96|13.12|20.98|12.71
> > > LLaMa-1B|9.77|9.82|15.64|9.59
> > >
> > > **2.$\alpha$-std Analysis:** Across 350M, 1B, and 3B models, LLR consistently achieves lower PPL while simultaneously reducing $\alpha$-std compared to the Uniform baseline:
> > > Llama-350M|0.001|0.0008|0.0006|0.0004|0.0002
> > > -|-|-|-|-|-
> > > Uniform-ppl|12.96|12.89|13.19|13.62|14.56
> > > LLR-ppl|12.71|12.56|12.75|13.27|13.57
> > > Uniform-std of alpha|0.30|0.29|0.28|0.30|0.32
> > > LLR-std of alpha|0.28|0.27|0.25|0.29|0.30
> > >
> > > LLaMA-1B and 3B:
> > >
> > > | | Uniform-ppl | LLR-ppl | Uniform-std of alpha | LLR-std of alpha |
> > > |-|-|-|-|-|
> > > | Llama-1B | 9.77 | 9.6 | 0.28 | 0.25 |
> > > | Llama-3B | 9.02 | 8.86 | 0.29 | 0.26 |
> > >
> > > We will include these scaled-up validations in the revised manuscript.
> > >
> > > >***Q2. The new LLaMA-3B pretraining and Llama-3.2-1B-Instruct fine-tuning results are good extensions, but they remain somewhat limited: the 3B run uses only 30B tokens, and the fine-tuning evaluation covers commonsense tasks only — my original question about instruction or reasoning workloads is still open.**
> > >
> > > ***A2:** To address concerns regarding the training horizon and the lack of reasoning evaluations, we conducted two extended experiments:
> > >
> > > **（1）Continued pretraining (LLaMA-3B):** We continued pretraining the 3B model for an additional 20K steps to verify LLR's efficacy beyond the initial 30B tokens.
> > >
> > > Method|Valid-PPL|OBQA|Winogran|ARC-C|ARC-E|Hellaswag|SIQA|PIQA|Lambada|Commonsense_QA|Avg.
> > > -|-|-|-|-|-|-|-|-|-|-|-
> > > Uniform|8.85|29.4|58.45|35.16|71.36|43.48|40.28|71.73|42.82|19.63|45.81
> > > LLR|8.71|30.6|60.28|36.49|72.64|44.29|41.62|72.81|44.6|20.86|47.13
> > >
> > > **（2）Reasoning Workloads:** To answer the open question regarding reasoning workloads, we evaluated the fine-tuned models on rigorous reasoning benchmarks, notably GSM8K.
> > >
> > > Llama-3.2-1B-Instruct|GSM8K|OBQA|ARC-C|Avg.
> > > -|-|-|-|-
> > > Un-tuned|32.90|24.60|35.75|31.08
> > > Uniform|34.80|25.00|38.82|32.87
> > > LLR|36.40|25.40|38.64|33.48
> > >
> > > >***Q3. The normalized LR distance analysis clearly quantifies the tuning overhead, but it was only computed on the smaller models (60M, 130M, GPT-nano). Showing this holds at 1B or 3B would be more convincing.**
> > >
> > > ***A3:** We are happy to provide this extended analysis at larger scales. Optimal AdamW LRs for `LLaMa-350M/1B/3B: 0.001/0.0005/0.0003`. The normalized distances are as follows:
> > >
> > > LR Dist|LAMB|LARS|Sharpness|LLR
> > > -|-|-|-|-
> > > **LLaMa-350M**|0.5|0|0.5|0
> > > **LLaMa-1B**|1|0.1|0.25|0
> > > **LLaMa-3B**|0.83|0.08|0.3|0
> > >
> > > We hope this resolves your concerns and welcome any further discussion. Your support and recognition mean a lot to us.

---

### Official Review · Reviewer_CjNq · 2026-03-10

**Soundness:** 2
**Presentation:** 3
**Significance:** 2
**Originality:** 2
**Overall Recommendation:** 4
**Confidence:** 3

**Summary:**

The paper proposes LLR, an adaptive per-layer learning rate scheme for LLM pre-training grounded in Heavy-Tailed Self-Regularization (HT-SR) theory. LLR periodically computes the ESD of each layer's weight correlation matrix, fits a power law to obtain PL_Alpha_Hill, and maps this to per-layer learning rates via a bounded linear scaling function. Layers with weaker heavy-tailedness get larger LRs; stronger heavy-tailedness gets smaller LRs. Three design choices are introduced: capping embedding LR, a soft switch to avoid LR spikes, and restricting updates to the first 20% of training. Experiments on LLaMA (60M-1B) and GPT-nano with AdamW and Muon show consistent perplexity improvements over uniform LR and existing layerwise methods, up to 1.5x speedup.

The HT-SR grounding is an interesting idea and empirical results within the chosen baselines are consistent. However, the absence of comparison against μP, which assigns per-layer learning rates from first principles with theoretical guarantees, is a critical gap that undermines the central claim. The correlational theoretical justification and the ad-hoc scaling function further weaken the work. I would encourage resubmission with μP/CompleteP comparisons.

**Compliance With Llm Reviewing Policy:**

Affirmed.

**Final Justification:**

Technically correct but limited in scope and evaluation, I have increased my score to weak accept following the rebuttal.

**Key Questions For Authors:**

1. How does LLR compare against μP and CompleteP? This comparison is essential to contextualize the contribution.

2. Can you disentangle whether PL_Alpha_Hill differences reflect training quality or architectural properties?

**Limitations:**

The paper does not discuss the relationship to μP-style parameterizations or the fragility of spectral estimates at small matrix sizes.

**Strengths And Weaknesses:**

**Strengths:**

The connection between HT-SR theory and layerwise LR assignment is well-motivated and distinct from prior heuristics like LARS or sharpness-based grid search.

The evaluation covers multiple architectures, scales, optimizers, and both pre-training and finetuning. Consistent improvements across the chosen baselines are convincing.


**Weaknesses:**

Missing Comparison Against μP: The paper does not compare against Maximal Update Parameterization (μP) (Yang et al., "Tensor Programs V") or extensions like CompleteP. μP assigns width-dependent per-layer learning rates from first principles to ensure stable feature learning at any scale. This is a principled layerwise LR scheme with theoretical guarantees. Given that LLR's central claim is that uniform LR is suboptimal and per-layer assignment helps, the absence of μP as a baseline is a critical gap. Without this comparison, it is impossible to assess whether LLR provides gains beyond what principled parameterization already achieves.

Correlational, Not Causal, Grounding: HT-SR theory establishes that well-trained networks exhibit heavier-tailed ESDs. However, some layers may be inherently less heavy-tailed due to architectural role (attention vs. FFN) rather than being undertrained. The paper observes this pattern but does not disentangle the two effects.

Ad-Hoc Scaling Function: The mapping (Equation 3) is a linear rescaling. No justification is given for linearity or this specific choice.

---

> ### Author Rebuttal · Authors · 2026-03-31
>
> We thank Reviewer for the thoughtful review and encouraging feedback.
>
> >**W1 & Q1. Missing $\mu P$ & CompleteP Baselines.**
>
> **A1:** Conceptually, $\mu P$ and CompleteP prioritize hyperparameter transfer stability across scales rather than explicitly maximizing absolute layer-wise performance. To compare comprehensively, we first grid-searched base LRs on LLaMA-135M:
> Llama-135M|0.005|0.001|0.0005|0.0001
> -|-|-|-|-
> **Uniform**|70.81|17.86|19.47|33.74
> **Mup-AdamW**|51.22|18.66|18.46|32.88
> **CompleteP-AdamW**|42.06|17.95|17.85|29.64
> **LLR**|**18.44**|**17.03**|**17.55**|**21.90**
>
> Using optimal LRs (1e-3 for Uniform/LLR; 5e-4 for $\mu P$/CompleteP), we evaluated across scales:
> Modelsize|Uniform|Mup-AdamW|CompleteP-AdamW|LLR|LLR with Mup-AdamW
> -|-|-|-|-|-
> **60M**|21.94|24.52|24.06|**20.3**|22.54
> **130M**|17.86|18.46|17.85|**17.03**|17.54
> **350M**|12.96|13.56|13.42|**12.71**|13.28
>
> LLR consistently outperforms both baselines. Furthermore, LLR+$\mu P$ yields strictly better performance than standard $\mu P$, confirming our dynamic spectral allocation provides orthogonal gains beyond static principled parameterizations.
>
> >**W2 & Q2. Disentangling architectural properties vs. training quality.**
>
> **A2:** Spectral disparities indeed fundamentally stem from architectural properties (Attention naturally exhibits heavier tails than FFNs)—recognizing this inherent heterogeneity is our core motivation. To rigorously disentangle structural priors from training dynamics, we evaluated an **Attention-only architecture**:
> Only-Att|0.001|0.0008|0.0005|0.0003
> -|-|-|-|-
> **Uniform**|22.80|23.23|24.8|27.98
> **LLR**|**22.23**|**22.31**|**23.27**|**24.62**
>
> These results demonstrate that LLR consistently outperforms the Uniform baseline even without architectural heterogeneity. This substantiates that LLR not only accommodates intrinsic structural differences (our primary motivation) but also optimizes training dynamics within homogeneous layers. We will incorporate this disentanglement analysis in the revised manuscript.
>
> >**W3. Ad-Hoc Scaling Function (Linearity).**
>
> **A3:** Thank you for pointing this out. We chose the linear mapping primarily for its simplicity and effectiveness. However, our method is highly robust to the choice of the scaling function. To demonstrate this, we compared our linear mapping with non-linear alternatives (Sqrt and Log2):
> - Sqrt: $f_t(i) = \eta \frac{\sqrt{\alpha_t^i}}{\frac{1}{L} \sum_{j=1}^{L} \sqrt{\alpha_t^j}}$
>
> - Log2: $f_t(i) = \eta \frac{\log_2(\alpha_t^i)}{\frac{1}{L} \sum_{j=1}^{L} \log_2(\alpha_t^j)}$
>
> The validation perplexity results are as follows:
> LR|Uniform|Linear (Ours)|Sqrt|Log2
> -|-|-|-|-
> **0.001**|17.86|**17.02**|17.16|17.14
> **0.0005**|19.47|**17.55**|17.94|17.86
>
> As shown, all variants (Linear, Sqrt, and Log2) significantly outperform the Uniform baseline. This proves that the performance gain comes from leveraging spectral information rather than a specific ad-hoc formula. We will add these ablation results to the appendix.
>
> >**Lim 1. Relation to $\mu P$ and fragility at small matrix sizes.**
>
> **A4:** For $\mu P$ comparisons, please refer to **A1**. Regarding small matrix fragility, we evaluated extremely small-scale models (LR=1e-3):
> Modelsize|Uniform|LLR
> -|-|-
> **40M**|24.54|**22.92**
> **20M**|38.26|**34.66**
>
> LLR consistently outperforms Uniform, demonstrating our spectral estimation remains highly robust and effective even at extremely small matrix scales.

---

> > ### Author Rebuttal · Reviewer_CjNq · 2026-03-31
> >
> > The authors have provided the primary comparisons I requested and showed substantial gains + composability with previous methods. With these additions, I generally consider this paper to be technically correct but limited in scope and evaluation. I shall increase my score accordingly

---

> > > ### Author Response · Authors · 2026-04-01
> > >
> > > Thank you very much for your positive feedback and for taking the time to review our work. We greatly appreciate your recognition and are glad that the additional experiment addressed your concerns. Your support is very important to us.

---

### Official Review · Reviewer_8K1q · 2026-03-11

**Soundness:** 3
**Presentation:** 3
**Significance:** 2
**Originality:** 2
**Overall Recommendation:** 2
**Confidence:** 4

**Summary:**

The manuscript claims to study the challenge of whether a single global learning rate is suboptimal for Transformer-based LLM training because different layers exhibit different spectral and optimization characteristics. The authors propose Layerwise Learning Rate (LLR), a dynamic layerwise schedule guided by heavy-tailed self-regularization theory: layers with weaker heavy-tailedness receive larger learning rates, while more heavy-tailed layers receive smaller ones. The method includes several Transformer-specific design choices, including special treatment for embeddings, a soft switching mechanism for layerwise LR updates, and an “active phase” that limits spectral updates to the early part of training. Empirically, the paper reports improved validation perplexity, faster convergence, and modest downstream gains across LLaMA-style models, GPT-nano, AdamW/Muon, and several commonsense benchmarks.

**Compliance With Llm Reviewing Policy:**

Affirmed.

**Key Questions For Authors:**

Refer to Weakness.

**Limitations:**

Refer to Weakness.

**Strengths And Weaknesses:**

**Strength**

The method is clearly specified, including the PL_Alpha_Hill metric, the bounded scaling rule, the soft-switch update, and the early active phase. The experimental section covers multiple model sizes (60M to 1B), multiple architectures, and two optimizers, and compares against several baselines including Uniform, LARS, LAMB, and a sharpness-based method. The main pretraining results are fairly consistent: LLR improves validation perplexity across all reported LLaMA scales, and the zero-shot commonsense results for the 1B model also improve on average. The additional ablations on fitting metrics, fitting frequency, and the scaling range (1,s) are helpful and increase confidence that the method is not overly brittle.

**Weakness**
1. A notable concern is that the empirical evaluation of model quality is still somewhat narrow for an LLM paper. The zero-shot evaluation focuses on seven commonsense multiple-choice tasks, and the fine-tuning results are again on a commonsense benchmark with RoBERTa-base, rather than on broader mainstream knowledge and reasoning evaluations such as MMLU, GSM8K, BBH, or math-oriented benchmarks. As a result, the paper supports the claim that LLR improves perplexity and some downstream transfer, but it is less conclusive about whether the method preserves or improves broader LLM capabilities.

2. A major weakness is that the theoretical justification remains rather coarse and not entirely convincing. In particular, the paper argues that layers with stronger heavy-tailedness are already better trained and therefore should receive smaller learning rates, whereas layers with weaker heavy-tailedness should receive larger ones. I find this interpretation insufficiently justified. Stronger heavy-tailedness does not necessarily imply that a layer is closer to an optimum or should be updated more conservatively. An alternative possibility is that such layers may lie on a favorable optimization manifold or encode more informative directions, in which case increasing, rather than decreasing, the learning rate could still be beneficial. More broadly, the paper currently presents this relationship as if it were mechanistically grounded, but the evidence seems largely correlational.

3. A notable weakness is the lack of discussion of key related work, such as [1], which studies layer-wise learning rates with finer-grained search and appears to arrive at a conclusion different from that of this paper. This weakens both the novelty claim and the theoretical motivation, because the manuscript currently presents its design principle somewhat in isolation, without adequately explaining why prior results point in a different direction.

[1].One LR Doesn’t Fit All: Heavy-Tail Guided Layerwise Learning Rates for LLMs

---

> ### Author Rebuttal · Authors · 2026-03-31
>
> We have carefully reviewed the points raised by Reviewer and address them point-by-point below.
>
> >**W1. Narrow empirical evaluation (missing MMLU/GSM8K/BBH, larger scale, modern LLM fine-tuning).**
>
> **A1:** We significantly expanded our experiments to address this:
>
> **(1) Scaling Up (LLaMA-3B on 30B Tokens LR=3e-4):**
> Method|Valid-PPL|OBQA|Winogran|ARC-C|ARC-E|Hellaswag|SIQA|PIQA|Avg.
> -|-|-|-|-|-|-|-|-|-
> Uniform|9.02|27.4|55.56|33.36|70.16|42.31|39.92|71.33|48.58
> LLR|**8.86**|**29.4**|**59.04**|**34.39**|**72.43**|**43.81**|**42.68**|**72.52**|**50.61**
>
> **(2) Preserving Broader Capabilities:** We evaluated `the above 3B models` on mainstream reasoning benchmarks:
> Method|Lambada|Commonsense_QA|COQA|MMLU|GSM8K|BBH
> -|-|-|-|-|-|-
> Uniform|42.54|19.41|2.51|25.2|0.2|26.5
> LLR|**44.77**|**20.39**|**2.54**|**25.4**|**0.5**|**27.8**
>
> *(Note: Scores are naturally low given the limited pre-training and lack of instruction tuning, but LLR consistently improves them).*
>
> **(3) Fine-Tuning Modern LLMs:** We fine-tuned `Llama-3.2-1B-Instruct` on `commonsense170k` (LR=3e-5, Batchsize=64, Epochs=3):
>
> Llama-3.2-1B-Instruct|OBQA|Winogran|ARC-C|ARC-E|Hellaswag|SIQA|PIQA|Commonsense_QA|Avg.
> -|-|-|-|-|-|-|-|-|-
> Un-tuned|24.60|59.91|35.75|68.31|44.98|41.66|74.37|55.20|49.94
> Uniform|25.00|64.80|**38.82**|68.14|45.29|**47.54**|72.25|61.67|52.94
> LLR|**25.40**|**65.00**|38.64|**68.64**|**45.49**|46.54|**76.25**|**62.67**|**53.58**
>
> These results confirm LLR safely preserves broader capabilities and transfers effectively to modern LLMs.
>
> >**W2. Weak theoretical justification. Why do heavy-tailed layers (smaller $\alpha$) need smaller LRs? Could they benefit from larger LRs?**
>
> **A2:** Our strategy—assigning smaller LRs to layers with smaller $\alpha$ (stronger heavy-tailedness)—is firmly supported empirically and theoretically:
>
> **1. Empirical Validation (LLR-rev):** To test your hypothesis that heavy-tailed layers might benefit from larger LRs, we tested an inverted mapping (LLR-rev) on LLaMA-130M, assigning larger LRs to layers with smaller $\alpha$.
>
> LR|Uniform|Random|LLR-rev|LLR
> -|-|-|-|-
> 0.001|17.86|17.97|27.18|**17.03**
> 0.0005|19.47|19.14|22.46|**17.55**
>
> Performance ranks: **LLR > Uniform $\gg$ LLR-rev**. Assigning larger LRs to heavy-tailed layers drastically degrades optimization (PPL jumps from 17.86 to 27.18).
>
> **2. Theoretical Justification:** The Mechanistic Link Between PL_Alpha_Hill $\alpha$ and LR $\eta$
>
> The catastrophic failure of LLR-rev can be mechanistically explained through the lens of Heavy-Tailed Self-Regularization (HT-SR) and Stochastic Differential Equations (SDEs). The core mechanism follows a clear causal chain: a smaller $\alpha$ leads to a lower intrinsic dimension, which strictly requires a smaller LR.
>
> *   **Small $\alpha \implies$ Lower Intrinsic Dimension:** As rigorously established by [1], a smaller tail-index $\alpha$ fundamentally restricts the optimization trajectory to a lower Hausdorff dimension ($d_H \downarrow$ ). This reduced intrinsic dimension $d_H$ acts as a strong implicit regularizer, **with Theorem 1 in [1] mathematically guaranteeing a tighter generalization bound** $E_{gen} \le \tilde{O}(\sqrt{d_H}) \text{, where } E_{gen} \text{ denotes the generalization error. In short, a small } \alpha$ indicates that the layer has successfully condensed its learned features into a highly generalizable, low-dimensional subspace.
> *   **Why a flat manifold requires a smaller LR:** From the well-established SDE perspective of SGD, once the layer has reached this low-dimensional subspace, maintaining a large LR severely degrades generalization[2,3]. In such a sensitive state, a large LR injects excessive noise variance, violently ejecting the layer from its well-generalized state (exactly explaining LLR-rev's failure).
>
> **Conclusion:** Therefore, layers with a small $\alpha$ require a decayed LR to maintain stability, whereas layers with a larger $\alpha$ need larger LRs to escape sharp minima. In short, the smaller the $\alpha$, the smaller the LR.
>
> [1] Simsekli, et al. Hausdorff dimension, heavy tails, and generalization in neural networks. NeurIPS 2020.
>
> [2] Dandi, et al. A random matrix theory perspective on the spectrum of learned features and asymptotic generalization capabilities. arXiv 2024.
>
> [3] Defilippis, et al. Scaling laws and spectra of shallow neural networks in the feature learning regime. arXiv 2025.
>
> >**W3. Missing related work [1] which arrives at a different conclusion.**
>
> **A3:** We carefully checked your provided citation [1] and noticed **it actually refers to our own submitted manuscript**. We conducted an extensive literature search and found no other prior work matching this exact title or description. Since our current submission cannot contradict itself as prior work, we suspect this might be a typo. If you were referring to a different paper, please provide the title/authors, and we will gladly discuss it.

---

> > ### Author Rebuttal · Reviewer_8K1q · 2026-04-05
> >
> > I am grateful for the authors' comprehensive and detailed rebuttal. I keep my score. I wish you all the best in your academic endeavors.

---

> > > ### Author Response · Authors · 2026-04-05
> > >
> > > Thank you for the reviewer’s follow-up. We appreciate the acknowledgment of our detailed rebuttal. However, the reviewer's response does not indicate which of our clarifications were unconvincing, nor does it engage with the concrete evidence we provided to address the original concerns. Maintaining the same score is of course the reviewer’s prerogative, but without any substantive justification, it is hard for us and the AC to assess whether the remaining concerns are still technical, or simply unaddressed. We provided detailed, point-by-point clarifications and evidence in the rebuttal, and we respectfully ask that the paper be evaluated on the basis of those technical issues and responses.
> > >
> > > All the best,
> > >
> > > Authors

---

### Official Review · Reviewer_Ssub · 2026-03-11

**Soundness:** 2
**Presentation:** 3
**Significance:** 3
**Originality:** 2
**Overall Recommendation:** 4
**Confidence:** 3

**Summary:**

This paper proposes Layerwise Learning Rate (LLR), a method that assigns different learning rates to each layer of a transformer during LLM pre-training. The approach is grounded in Heavy-Tailed Self-Regularization (HT-SR) theory: the authors compute the empirical spectral density (ESD) of each layer's weight correlation matrix, fit a power law, and use the resulting PL_Alpha_Hill exponent to determine per-layer learning rates. Layers with higher PL_Alpha_Hill (weaker heavy-tailedness, e.g., embeddings and FFN layers) receive larger learning rates, while layers with lower values (stronger heavy-tailedness, e.g., attention layers) receive smaller ones. Experiments span LLaMA (60M–1B) and GPT-nano architecture (135M) with AdamW and Muon optimizers, reporting perplexity improvements, and zero-shot accuracy gains (47.09% → 49.02% on LLaMA-1B)

**Compliance With Llm Reviewing Policy:**

Affirmed.

**Final Justification:**

The paper remains strong, and the addition of missing baselines effectively addressed one of my main critiques. That said, the investigation into the Soft switch mechanism remains incomplete. The rebuttal experiments for phases beyond warmup used a different number of steps than the original setup, which obscures the results. I suggest the authors elaborate further on the relationship between the training stage and the optimal step count required for the Soft Switch mechanism. Because the primary concerns were addressed despite this new concern, I maintain my original assessment.

**Key Questions For Authors:**

1. Warmup vs. LLR Phase: Why was the decision made to overlap the LLR measurement phase with the learning rate warmup phase? Wouldn't it be more theoretically sound (and empirically stable) to delay the start of the LLR active phase until after the 10% warmup is complete (e.g., running LLR from 10% to 30% of training tokens)?

2. 20% Cutoff Across Scales: Does the 20% active phase cutoff hold true for the 1B parameter model? If not, how would the computation overhead scales for larger models?

3. Measurement Frequency: Appendix B (Figure 9) shows that a PL fitting gap of 100 steps works well. Is this 100-step gap a static hyperparameter, or should it be a dynamic ratio based on the total batch size?

4. Adam-mini Comparison: Can the authors provide a performance comparison between LLR and Adam-mini?

**Limitations:**

- As explicitly noted by the authors, the approach has not been evaluated on multimodal tasks (e.g., vision-language models), where the optimization dynamics of different modalities might introduce competing spectral behaviors.

- The evaluation ceiling is restricted to 1 billion parameters, leaving its efficacy on cutting-edge, large-scale models untested.

**Strengths And Weaknesses:**

Strengths:

- Clear presentation: The method is presented very nicely with clear experimental results.

- Consistent Empirical Gains: The method consistently outperforms uniform baselines across varying scales (60M to 1B parameters) and across different optimizers (AdamW and Muon).

- Low Tuning Overhead: Unlike Sharpness or LAMB, LLR can inherit its base parameters directly from an established, near-optimal uniform learning rate setup without exhaustive re-tuning.

Weaknesses:

- Missing Baseline (Adam-mini): The authors compare LLR against LARS and LAMB, which are older methods. While the authors appropriately cite Adam-mini (Zhang et al., 2024b) as a recent layer/block-wise learning rate allocation method in their Related Work, they omit it from their empirical evaluation.

- Warmup Overlap: The method restricts layer-wise LR updates to the first 20% of training tokens (the "Active Phase"). However, the experimental setup notes that the first 10% of training tokens are used for learning rate warmup. This means half of the LLR active phase overlaps with the warmup phase. HT-SR theory relies on measuring heavy-tailedness in "well-trained models". Measuring spectral density during the highly chaotic warmup phase seems theoretically conflicting and prone to capturing initialization artifacts.

- Unverified Cross-Scale Generalization of Heuristics: The authors claim that LLR statistics stabilize after the first 20% of tokens, justifying the "Efficient Active Phase" cutoff. However, the ablation study proving this (Figure 4, Right) is only conducted on the 135M parameter model. It is not clear whether this 20% threshold successfully scales to the 1B parameter model, or if larger models require a longer active phase to properly stabilize their spectral densities.

- Reliance on Chinchilla-Optimal Scaling vs. Modern Overtraining: The pre-training experiments are strictly conducted under the Chinchilla scaling law. However, modern LLM pre-training paradigms heavily favor overtraining models well beyond compute-optimal token limits to achieve better inference efficiency. It is unclear how the PL_Alpha_Hill dynamics evolve in an overtrained regime.

---

> ### Author Rebuttal · Authors · 2026-03-31
>
> We sincerely thank Reviewer for the constructive feedback and valuable suggestions.
>
> >**W1 & Q4. Missing Adam-mini baseline.**
>
> **A1:** We added Adam-mini across 60M, 130M, and 350M models. We grid-searched its LR in `{1e-3, 5e-4, 1e-4}` (optimal consistently 1e-3).
>
> Method|60M|130M|350M
> -|-|-|-
> **Uniform**|21.94|17.86|12.96
> **Adammini**|21.85|17.72|13.04
> **LLR**|**20.30**|**17.03**|**12.71**
>
> LLR consistently outperforms Adam-mini, confirming our dynamic strategy's superiority.
>
> >**W2 & Q1. Warmup Overlap (0-20% vs 10-30%).**
>
> **A2:** Overlapping LLR with warmup is driven by:
> 1) **Theoretical:** Capturing Early Dynamics. While HT-SR often analyzes well-trained models, prior work [1] has demonstrated that distinct, non-random layer-wise spectral differences emerge almost immediately after the first few gradient steps. Our method intentionally leverages these early structural differences to guide the optimization trajectory from the very beginning, rather than waiting for the model to stabilize.
> 2) **Empirical:** Starting LLR after warmup (e.g., at 10%) causes abrupt "LR shocks" and unrecoverable loss spikes.
>
> Modelsize|Uniform|0%-100%|0%-20%(ours)|10%-30%|20%-40%|20%-100%
> -|-|-|-|-|-|-
> **60M**|21.94|**20.31**|20.35|38.50|56.99|52.10
> **130M**|17.86|**17.00**|17.06|28.15|38.46|34.46
> **350M**|12.96|**12.67**|12.71|19.54|28.51|26.24
>
> Delaying LLR (10-30%) degrades performance. Our 0-20% setting matches full-duration performance efficiently.
>
> [1] Jimmy Ba, et al. High dimensional asymptotics of feature learning: How one gradient step improves the representation.
>
> >**W3 & Q2. Cross-Scale Generalization (20% Cutoff).**
>
> **A3:** To verify this cross-scale generalization, we evaluated the 1B model (trained on 20B tokens) comparing the AdamW baseline, our efficient `LLR-20%` (20% active phase), and `LLR-100%` (full active phase):
>
> Method|PPL|OBQA|Winogrande|ARC-c|ARC-e|Hellaswag|SIQA|PIQA|Avg.
> -|-|-|-|-|-|-|-|-|-
> **adamw**|9.77|28.0|55.01|30.72|66.50|39.29|40.38|69.75|47.09
> **LLR-20%**|9.595|**29.6**|56.67|**34.64**|**70.46**|**40.53**|**40.79**|70.46|**49.02**
> **LLR-100%**|**9.589**|29.2|**56.88**|34.35|69.90|40.45|40.48|**71.27**|48.93
>
> `LLR-20%` matches `LLR-100%` and beats AdamW. Since larger models don't require prolonged active phases, overhead remains strictly bounded to early training.
>
> >**W4. Chinchilla Scaling vs. Overtraining.**
>
> **A4:** To test overtraining, we extended 1B pre-training to `100B tokens`.
>
> Method|Train-loss|Valid-PPL|OBQA|Winogran|ARC-C|ARC-E|hellaswag|siqa|piqa|Avg.
> -|-|-|-|-|-|-|-|-|-|-
> **AdamW**|**2.181**|**8.912**|30.8|57.93|37.37|**73.40**|44.48|41.71|72.42|51.16
> **LLR**|2.184|8.935|**32.2**|**60.38**|**38.40**|72.35|**45.24**|**41.80**|**73.01**|**51.91**
>
> While loss/PPL converge, LLR achieves higher downstream accuracy (51.91 vs 51.16). AdamW slightly overfits the pre-training objective, whereas LLR balances spectral features for better true downstream generalization.
>
> >**Q3. Measurement Frequency (100-step gap).**
>
> **A7:** This is a **token-based dynamic interval**. A 100-step gap corresponds to ~5.2M tokens. If batch size scales, steps should reduce proportionally. The hyperparameter is highly robust: varying the gap from 50 to 500 steps (on 135M) only fluctuates PPL by $\le$ 0.15.
>
> >**Lim 1. Multimodal task evaluation.**
>
> **A8:** We trained a Vision Transformer (ViT-Tiny) from scratch on ImageNet-1K:
>
> Backbone/Dataset|Metric|Uniform|LLR
> -|-|-|-
> **ViT-Tiny/Imagenet-1K**|Top-1 $\uparrow$ |67.42%|**68.51%**
>
> The +1.09% improvement confirms LLR's spectral balancing generalizes to vision, establishing a solid foundation for future VLM training. We will add these results to the revised appendix.
>
> >**Lim 2. Scale restricted to 1B parameters.**
>
> **A9:** We trained a **LLaMA-3B** model on 30B tokens (LR=3e-4) to validate scalability:
>
> LLaMa-3B|Valid-PPL|OBQA|Winogran|ARC-C|ARC-E|Hellaswag|SIQA|PIQA|Avg.
> -|-|-|-|-|-|-|-|-|-
> **Uniform**|9.02|27.4|55.56|33.36|70.16|42.31|39.92|71.33|48.58
> **LLR**|**8.86**|**29.4**|**59.04**|**34.39**|**72.43**|**43.81**|**42.68**|**72.52**|**50.61**
>
> LLR consistently outperforms Uniform at the 3B scale, reducing PPL by 0.16 and boosting zero-shot Avg by +2.03.

---

> > ### Author Rebuttal · Reviewer_Ssub · 2026-04-01
> >
> > Thank you to the authors for the detailed rebuttal. I am convinced by your responses to most of my questions and appreciate the additional context.
> >
> > Regarding the "Warmup Overlap," while your theoretical point about capturing early spectral dynamics is fair, your empirical explanation reveals a contradiction. You note that starting LLR after the 10% warmup causes "unrecoverable loss spikes." However, the paper explicitly introduces the "Soft Switch" mechanism to solve this exact problem, claiming it "eliminates spikes, thereby enhancing training stability".
> >
> > If the Soft Switch fails to prevent these shocks post-warmup, this is a limitation that needs to be explicitly addressed. It implies LLR functions primarily as an early-stage trajectory setter rather than a fully robust dynamic scheduler. This distinction is practically important: in use cases like continued pretraining or domain adaptation, practitioners need stable layer-wise adjustments mid-training and cannot rely on the extreme malleability of a from-scratch warmup phase.
> >
> > I will maintain my current score for now.

---

> > > ### Author Response · Authors · 2026-04-03
> > >
> > > We sincerely thank the reviewer for the continued engagement and for raising this highly insightful question. We apologize if our previous brevity caused a misunderstanding. We clarify that LLR is a robust dynamic scheduler, and the previously observed "unrecoverable loss spikes" were simply caused by **an overly short transition window applied to a massive LR gap**.
> > >
> > > >***Q1.If the "Soft Switch" eliminates loss spikes as claimed, why does starting LLR post-warmup cause unrecoverable shocks?**
> > >
> > > ***A1:** Here is the detailed clarification and new empirical evidence:
> > >
> > > *   **Original `Soft Switch`:** It was designed for continuous, step-to-step layerwise LR adjustments. For standard small shifts (e.g., global `LR from 3.5e-4 to 3.6e-4`), a **50-step window** creates a gentle per-step change ($\Delta LR \approx 2\times 10^{-7}$), eliminating spikes.
> > > *   **The Cause of Spikes:** However, initiating LLR *after* a 10% warmup requires a massive LR jump (e.g., setting $s=5$ of LLR, `LR from 1e-4 to 5e-4`). In our previous 10%-30% ablation, we **mistakenly kept the same 50-step window for this huge gap**. This forced a per-step change of $\Delta LR \approx 2\times 10^{-6}$—10 times larger than our original intended setting.
> > > *   **Solution:** To fix this, we simply scaled **the `Soft Switch` window to 500 steps** for the delayed LLR (10%-30%). As shown below, post-warmup LLR configurations now train perfectly stably and outperform the Uniform baseline:
> > >
> > > Modelsize|Uniform|0%-100%|0%-20%(ours)|10%-30%|20%-40%|20%-100%
> > > -|-|-|-|-|-|-
> > > **60M**|21.94|**20.31**|20.35|20.58|21.15|21.06
> > > **130M**|17.86|**17.00**|17.06|17.32|17.49|17.38
> > > **350M**|12.96|**12.67**|12.71|12.79|12.86|12.82
> > >
> > > **Conclusion:** With a proper transition window, LLR initiates safely at any stage. Yet, starting at 0-20% remains the most cost-effective strategy. We will include this ablation in the revision.
> > >
> > > >***Q2. This distinction is practically important: in use cases like continued pretraining or domain adaptation, practitioners need stable layer-wise adjustments mid-training and cannot rely on the extreme malleability of a from-scratch warmup phase.".**
> > >
> > > ***A2:** To demonstrate that LLR is robust in these scenarios and does not rely on a from-scratch warmup, we conducted two new experiments:
> > >
> > > **(1) Continued Pretraining**: Resuming from the 3B-30B checkpoint (see **A9**), we trained for another 10k steps with a strictly fixed learning rate of 3e-5.
> > >
> > > Method|Valid-PPL|OBQA|Winogran|ARC-C|ARC-E|Hellaswag|SIQA|PIQA|Lambada|Commonsense_QA|Avg.
> > > -|-|-|-|-|-|-|-|-|-|-|-
> > > Uniform|8.92|27.8|55.74|33.42|70.22|42.42|39.95|71.48|42.56|19.45|44.78
> > > LLR|8.77|29.6|59.24|34.48|72.18|43.81|42.76|72.62|44.8|20.46|46.66
> > >
> > > **(2) Domain Adaptation (Fine-Tuning):** We fine-tuned `Llama-3.2-1B-Instruct` on `Commonsense170k` (LR=3e-5, batchsize=64,Epochs=3)
> > >
> > > | Llama-3.2-1B | OBQA | Wino. | ARC-c | ARC-e | Hella. | SIQA | PIQA | CSQA | Avg. |
> > > | :--- | :--- | :--- | :--- | :--- | :--- | :--- | :--- | :--- | :--- |
> > > | Un-tuned | 24.60 | 59.91 | 35.75 | 68.31 | 44.98 | 41.66 | 74.37 | 55.20 | 49.94 |
> > > | Uniform | 25.00 | 64.80 | **38.82** | 68.14 | 45.29 | **47.54** | 72.25 | 61.67 | 52.94 |
> > > | **LLR** | **25.40** | **65.00** | 38.64 | **68.64** | **45.49** | 46.54 | **76.25** | **62.67** | **53.58** |
> > >
> > > These results confirm LLR safely preserves broader capabilities.
> > >
> > > We sincerely hope our detailed replies have fully addressed your concerns. We would be delighted to continue the discussion if any questions remain, as your support and endorsement are incredibly important to our work.

---

### Decision · Program_Chairs · 2026-04-30

**Decision:**

Accept (regular)

**Comment:**

The paper proposes layerwise learning rate allocation guided by heavy-tailed spectral statistics for LLM pre-training. Reviewers agreed the method is well-specified with consistent gains across scales (60M to 3B) and optimizers (AdamW, Muon). The rebuttal addressed key concerns: missing baselines (μP, Adam-mini, CompleteP) were added with LLR outperforming all, and an inverted-schedule ablation provided convincing causal evidence. Two minor issues remain: the Soft Switch window needs systematic characterization across training stages, and reasoning evaluation at the current scale is not yet convincing. The paper meets the acceptance bar as a practical contribution to LLM optimization.